# Brain-wide analysis of the supraspinal connectome reveals anatomical correlates to functional recovery after spinal injury

Zimei Wang[1], Adam Romanski[1], Vatsal Mehra[1], Yunfang Wang[2], Matthew Brannigan[1], Benjamin C Campbell[3], Gregory A Petsko[3], Pantelis Tsoulfas[2]*, Murray G Blackmore[1]*

[1]Department of Biomedical Sciences, Marquette University, Milwaukee, United States; [2]Department of Neurological Surgery, University of Miami, Miami, United States; [3]Helen and Robert Appel Alzheimer's Disease Research Institute, Cornell University, New York, United States

**Abstract** The supraspinal connectome is essential for normal behavior and homeostasis and consists of numerous sensory, motor, and autonomic projections from brain to spinal cord. Study of supraspinal control and its restoration after damage has focused mostly on a handful of major populations that carry motor commands, with only limited consideration of dozens more that provide autonomic or crucial motor modulation. Here, we assemble an experimental workflow to rapidly profile the entire supraspinal mesoconnectome in adult mice and disseminate the output in a web-based resource. Optimized viral labeling, 3D imaging, and registration to a mouse digital neuro-anatomical atlas assigned tens of thousands of supraspinal neurons to 69 identified regions. We demonstrate the ability of this approach to clarify essential points of topographic mapping between spinal levels, measure population-specific sensitivity to spinal injury, and test the relationships between region-specific neuronal sparing and variability in functional recovery. This work will spur progress by broadening understanding of essential but understudied supraspinal populations.

*For correspondence:
ptsoulfa@Med.miami.edu (PT);
murray.blackmore@marquette.
edu (MGB)

## Editor's evaluation

This work is of interest to neuroscientists interested in tissue clearing, viral labeling, and its applications to spinal cord injury in particular, but not exclusively. It provides a significant methodological advance applied for investigating descending pathways from a fundamental perspective and then in the context of potential recovery after a spinal cord injury. This study represents an important new direction and set of tools for the field of motor control.

## Introduction

The brain's control of the body below the head is achieved largely by axonal inputs to spinal circuits, which then interpret and process descending signals to generate appropriate commands to the periphery through motor and autonomic output neurons. The supraspinal connectome is highly conserved in mammals, with multiple cell types distributed through the brainstem, midbrain, and motor cortex, each projecting axons to a subset of spinal levels and to selected cell types (*Nudo and Masterton, 1988*; *Kuypers and Martin, 1982*). A comprehensive and accessible approach to understanding supraspinal input is crucial to interpret motor and autonomic behavior, and treat conditions that disrupt descending signals such as stroke, disease, or injury to the spinal cord.

Extensive work spanning more than a century has employed orthograde degeneration, electrical stimulation, and axonal transport-tracing methods to characterize the location and function of specific supraspinal neurons in various animals, providing a base of knowledge to understand supraspinal control (*Nudo and Masterton, 1988*; *Kuypers and Martin, 1982*; *Hoff, 1932*; *Glees, 1946*; *ten Donkelaar, 2000*). Several efforts in rodents have provided more global information by performing retrograde tracing from selected spinal levels, followed by tissue sectioning and manual assignment of labeled cell bodies to regions within the brain (*Lakke, 1997*; *Leong et al., 1984*; *Liang et al., 1997*). Significant challenges, however, impede the distribution of this foundational knowledge and its application to the study of disease and injury-based disruptions. First, information about the location and types of supraspinal neurons is fragmented and not standardized across numerous studies (*Tuszynski and Steward, 2012*). Second, a high level of expertise is required to precisely identify brain regions from two-dimensional (2D) tissue series and build a three-dimensional (3D) view of the connections (*Gong et al., 2013*; *Oh et al., 2014*; *Economo et al., 2016*). Third, tissue sectioning and imaging are laborious and time consuming, making experiments to track dynamic changes after injury or disease impractical. Consequently, attention has remained focused on a relatively narrow set of supraspinal populations. For example, in the field of spinal cord injury (SCI), the vast majority of studies concern only a handful of descending populations, notably the corticospinal, rubrospinal, raphespinal, and broadly defined reticulospinal (*Fink and Cafferty, 2016*; *Anderson, 2004*; *Kwon et al., 2002*; *Lu et al., 2012*; *Blackmore et al., 2021*; *Takakusaki et al., 2016*). This attention is justified as these regions serve important motor functions and comprise a majority of descending input (*Lemon, 2008*). On the other hand, dozens of additional brain regions also project to the spinal cord, many of which carry essential motor and autonomic commands (*Liang et al., 1997*). Without tools to easily monitor the totality of the supraspinal connectome, researchers lack even basic information regarding their sensitivity to injury, innate plasticity, or potentially disparate responses to potential pro-regenerative therapies.

Here, we present a comprehensive and accessible approach to obtain detailed information about the number and location of descending projection neurons throughout the mouse brain. By combining retrograde viral labeling (*Tervo et al., 2016*), 3D imaging of optically cleared brains (*Wang et al., 2018*), and registration to standard neuroanatomical space (*Niedworok et al., 2016*; *Tyson et al., 2021*), we rapidly identify the specific location of tens of thousands of supraspinal neurons. We present a web-based resource that compares the locations and quantity of supraspinal neurons that project to cervical versus lumbar levels. We further extend this approach to questions related to SCI by quantifying the region-specific sparing of distinct supraspinal populations in mice that received injuries of graded severity. These data provide an accessible resource to disseminate detailed understanding of the supraspinal connectome and provide an example of combined methods that achieve brain-wide profiling of brain–spinal cord connectivity after injury.

## Results

### Optimization of retrograde cell detection in cleared brain tissue

Although tissue-clearing techniques offer unprecedented visualization of intact neural structures, overall degradation of fluorescent signal during the process, compounded by optical interference from nearby axon tracts, can significantly limit detection of cell bodies (*Steward et al., 2021*; *Asboth et al., 2021*; *Frezel et al., 2020*). This detection problem is evident in prior experiments from our lab and others that used spinal injection of AAV2-retro to retrogradely express fluorescent proteins (FPs) in supraspinal neurons (*Wang et al., 2018*; *Steward et al., 2021*). Fluorescent intensity in these experiments was variable between different populations, and in dimmer regions only a small fraction of total supraspinal neurons were detected in 3D images. Detection of supraspinal neurons was especially low in the brainstem, a phenomenon also noted in a recent study (*Steward et al., 2021*). We therefore tested a strategy of nuclear localization to improve detection (*Sathyamurthy et al., 2020*). Indeed, 2 weeks after delivery of retrograde vectors to the lumbar spinal cord, detection of supraspinal neurons in cleared brain tissue was significantly enhanced when mScarlet (mSc) fluorophore (*Bindels et al., 2017*) was localized to the nucleus by an H2B tag (*Figure 1—figure supplement 1*). Spot detection confirmed an approximately 2-fold increase in the number of detected cells in the cortex and more than 10-fold increase in the brainstem (249.6 ± 56.3 SEM vs. 3334.3 ± 82.8 SEM, *Figure 1—figure*

*supplement 1I–L*). Close examination of spot detection of nuclear-localized label showed that even in areas of relatively densely packed supraspinal neurons the cell nuclei were well spaced, enabling effective segregation and detection (*Figure 1—figure supplement 1I–K*).

Next, to further optimize cell detection we tested mGreenLantern (mGL), a recently described green FP with enhanced brightness (*Campbell et al., 2020*). To directly compare mGL to mSc, adult mice received lumbar (L1-L2) injection of mixed AAV2-retro-H2B-mGL and -mSc, followed 2 weeks later by brain clearing, 3D imaging, and nuclei detection using Imaris software. Based on location ,we classified retrogradely labeled nuclei into six groups: corticospinal, hypothalamic, red nucleus (RN), dorsal pons, medullary reticular formation, and caudal dorsal medulla (*Figure 1—figure supplement 2A–N*). Note that additional nuclei existed outside these easily recognizable areas and are considered below. Quantification of labeled objects revealed that H2B-mGL significantly increased detection of neurons in cortex, dorsal pons, and reticular formation (p<0.01, two-way ANOVA with post-hoc Sidak's) (*Figure 1—figure supplement 2L–O*). Counts were similar in right and left hemispheres, confirming that spinal injections reached both sides equally (*Figure 1—figure supplement 3A*), and did not increase between 2 and 4 weeks post-injection (*Figure 1—figure supplement 3B*). Moreover, signal was readily detectable without the need for antibody amplification, thus avoiding lengthy incubation periods. A video further illustrates the improved detection of supraspinal neurons with nuclear-localized mGL (*Figure 1—video 1*). Combined, these data establish an initial categorization and quantification of supraspinal brain regions in 3D space, reveal H2B-mGL to be the most sensitive of the FPs tested, and create consistent experimental parameters for the detection of supraspinal neurons.

## A pipeline for the detection and spatial registration of supraspinal projection neurons

The procedures outlined above improve detection of supraspinal nuclei yet remain reliant on user-generated judgments regarding cell location. This is likely only practical and accurate for large and isolated populations, and new approaches are needed for closely adjacent or intermingled brain regions (*Tyson and Margrie, 2021*). We therefore established an analysis pipeline to standardize brain registration and cell detection using tools and concepts derived from the Brainglobe initiative (*Figure 1A–D*). After tissue clearing and imaging, image stacks were imported to Imaris software for visual quality control such as detection of autofluorescent artifacts or tissue damage. Image stacks were then exported and preprocessed to create cell and background sets in standard orientation (*Figure 1A and B*), followed by registration and segmentation using automated mouse atlas propagation (aMAP/brainreg), a well-validated tool to align 2D datasets with the 25 μm version of Allen Mouse Brain Atlas (*Niedworok et al., 2016*; *Tyson et al., 2021*; *Kim et al., 2015*; *Figure 1C*). We next used cellfinder, a deep learning model-based tool, for the identification of labeled cells in whole-brain images (*Tyson et al., 2021*; *Figure 1C*). In conjunction with aMAP/brainreg, cellfinder assigns objects to 645 individual brain regions and quantifies the number in each. In addition, cellfinder produces detailed visualization of each optical slice, with defined brain regions outlined and labeled cells represented as overlaid spots (*Figure 1C*, *Figure 1—figure supplement 4*). For final visualization, we used brainrender (*Claudi et al., 2021*), which displays cellfinder output in an interactive 3D format registered to the Allen Mouse Reference Atlas (*Figure 1D*). To disseminate the insights from this pipeline, both 2D and 3D visualizations of the brains described below are available on an interactive web interface (3Dmousebrain.com).

## Brain clearing and registration quantify supraspinal connectivity to the lower spinal cord

We first applied the registration pipeline to examine connectivity from the brain to the lower spinal cord. Ten animals received injection of AAV2-retro-H2B-mGL to L1 spinal cord, followed 2 weeks later by perfusion, imaging, and analysis. On average, 31,219 nuclei were detected per brain (range 20,688–40,171). Supraspinal nuclei were detected in 69 of the 645 total brain regions. To simplify reporting, we reduced this to 25 summary categories, for example, spinal, medial, and lateral vestibular nuclei were collapsed into a single vestibular category (*Figure 1—figure supplement 4D*). We use the 25-region summary counts to report findings throughout this article, while counts in both the original 69-region and 25-region summary format are supplied for all animals in *Source data 1*. Next

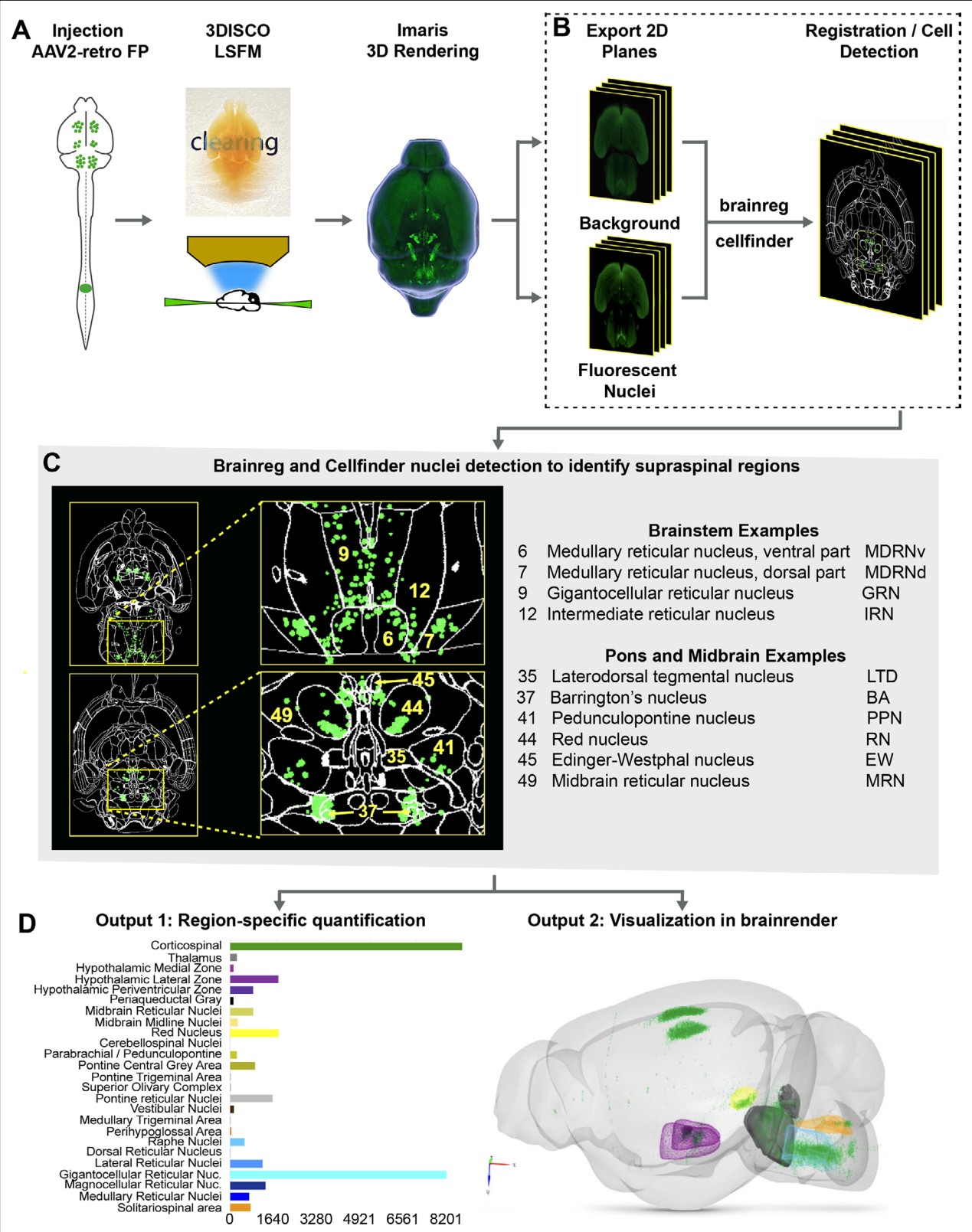

**Figure 1.** A pipeline and web resource for the detection and spatial registration of supraspinal projection neurons. (**A**) Tissue preparation and initial imaging. AAV2-retro-H2B-mGL is injected to the spinal cord, followed by perfusion, tissue clearing, light sheet imaging, and 3D processing using Imaris software. (**B**) Image registration. A complete series of background and fluorescent nuclei images are exported for registration to standard 3D space by brainreg and cell nuclei detection by cellfinder. (**C**) An example of cellfinder output, showing horizontal brain sections with brain regions outlined

*Figure 1 continued on next page*

*Figure 1 continued*

and detected cell nuclei indicated in green. (**D**) Example output available at https://3Dmousebrain.com/. On the left are quantitative nuclei counts for identified brain regions, and on the right is 3D visualization of supraspinal locations generated by brainrender, an interactive Python-based tool.

The online version of this article includes the following video and figure supplement(s) for figure 1:

**Figure supplement 1.** Nuclear localization of retrograde fluorophores enhances detection of supraspinal neurons in cleared brain tissue.

**Figure supplement 2.** Comparison of supraspinal labeling by nuclear-localized mScarlet and mGreenLantern.

**Figure supplement 3.** Detection of supraspinal neurons is similar across the midline and maximal by 2 weeks post-injection.

**Figure supplement 4.** 3D imaging followed by image registration and nuclei detection by cellfinder and brainreg identifies brain regions containing supraspinal projection neurons.

**Figure supplement 5.** Registration of mouse brains injected at T10, L1, and L3-4 shows a similar pattern of labeled supraspinal neurons.

**Figure 1—video 1.** A video illustrating the improvement in the detection of supraspinal cells by retrograde labeling using nuclear-localized mGreenLantern versus cytoplasmic mScarlet.

https://elifesciences.org/articles/76254/figures#fig1video1

we highlight the variety of supraspinal populations identified by 3D registration, with reference to prior descriptions in rodent that help validate the automated findings. Note that we adopt nomenclature from the Allen Mouse Reference Atlas, which the cellfinder pipeline employs.

In the medulla, the largest group of supraspinal neurons registered to the gigantocellular nucleus (GRN), an evolutionarily conserved source of both pre-autonomic and pre-motor axons (*Figure 1— figure supplement 4B8*; *Aicher et al., 1995*; *Brownstone and Chopek, 2018*; *Liang et al., 2016*; *Hermann et al., 2003* ). More ventrally, labeled nuclei were present in the magnocellular reticular nucleus (MARN), which also project to the ventral horn and IML of the lower spinal cord (*Figure 1— figure supplement 4B10*). Notably, labeled nuclei also mapped to regions lateral to the GRN, including the paragigantocellular reticular nucleus, lateral part (PGRNl) (*Figure 1—figure supplement 4B10*). This region contains spinally projecting neurons that may initiate locomotion, as well as the ventral rostral medullary group that regulates blood pressure (*Capelli et al., 2017*; *Van Bockstaele et al., 1989*; *Opris et al., 2019*). A cluster of labeled nuclei was also located dorsally in the caudal medulla, within and near the solitary nucleus, as supraspinal region implicated in visceral input to respiration and cardiovascular tone (*Leong et al., 1984*; *Liang et al., 1997*; *Mtui et al., 1995* ; *Figure 1—figure supplement 4B3*).

More rostrally in the brainstem, labeled nuclei were present in the spinal and medial vestibular nuclei, which project to spinal targets to mediate postural control (*Figure 1—figure supplement 4B3*). Labeled nuclei were also abundant in the pontine reticular nuclei (*Leong et al., 1984*; *Liang et al., 1997*; *Figure 1—figure supplement 4B6*). Although perhaps less well understood than medullary reticular populations, pontine reticular neurons have been linked to muscle atonia during sleep, startle responses, and to multisegment postural adjustments during limb extension (*Perreault and Giorgi, 2019*; *Takakusaki et al., 2016* ). More dorsally in the pons, labeled nuclei mapped to known supraspinal regions in and around the pontine central gray, including the locus coeruleus (LC), laterodorsal and sublaterodorsal tegmental nucleus (*Leong et al., 1984*; *Liang et al., 1997 Cornwall et al., 1990*; *Peever and Fuller, 2016*; *Sluka and Westlund, 1992*; *Figure 1—figure supplement 4B3*). Labeled nuclei also registered to Barrington's nucleus (BAR), which plays a central role in the control of micturition and bowel control (*Barrington, 1921 Verstegen et al., 2017*; *Figure 1—figure supplement 4B2*). Another prominent nucleus was the parabrachial nucleus (PBN), a sensory relay for inputs related to itch, pain, touch, and a range of autonomic controls including blood pressure and thermoregulation (*Figure 1—figure supplement 4B3*; *Chiang et al., 2019*; *Choi et al., 2020* ). Notably, although deep cerebellar nuclei (DCN) project to lumbar spinal cord and were recently shown to retrogradely express lumbar-injected AAV2-retro (*Sathyamurthy et al., 2020*), detection of cell nuclei in this region was low and highly variable between animals. Although visual inspection confirmed label in this region, albeit dim, the extreme dorsal position of the nuclei likely acted to lower detection by the ventrally positioned camera, such that the deep learning algorithm was not successful in distinguishing the residual signal from background. Accordingly, we considered the DCN values unreliable and do not include them in results from the automated pipeline.

In the midbrain, supraspinal nuclei were prominently detected in the RN, as expected, and in midline regions including Edinger–Westphal (EW) and the interstitial nucleus of Cajal (INC), whose

supraspinal projections are known to affect postural adjustments and energy homeostasis (*Figure 1—figure supplement 4B*; *Leong et al., 1984 Yu and Wang, 2020*; *Kozicz et al., 2011*). Interestingly, nuclei also mapped to the midbrain reticular nucleus and pedunculopontine nucleus (PPN) (*Figure 1—figure supplement 4B2*), which lie within the mesencephalic locomotor region (MLR), a well-studied region in which stimulation triggers locomotion in a range of species including mice (*Caggiano et al., 2018*; *Roseberry et al., 2016*). Although much MLR activity acts through reticular relays, the presence of direct supraspinal input from the PPN and mesencephalic reticular nucleus (MRN), noted here and elsewhere, indicates some role in direct spinal activation (*Caggiano et al., 2018*; *Gatto and Goulding, 2018*; *Dautan et al., 2021*; *Ferreira-Pinto et al., 2021*).

Finally, in the forebrain, large clusters of nuclei were detected in the corticospinal tract region, as expected. These are considered in more detail below. Supraspinal neurons were also detected in the hypothalamus, where they separated into two prominent clusters, medial and lateral (*Figure 1—figure supplement 4B7*). The medial cluster mapped mostly to the paraventricular hypothalamic nucleus (PVH) and the adjacent descending paraventricular nucleus (PVHd). These are known to innervate autonomic circuitry in the lower spinal cord and modulate functions including bladder control, sexual function, and blood pressure (*de Groat et al., 2015*; *Holstege, 2005*; *Zhou et al., 2019*). The lateral cluster spanned the dorsomedial nucleus (DMH) and the lateral hypothalamic area (LHA). Although less well characterized than the PVH, prior work in the LHA has identified orexin-expressing neurons that project to all spinal levels with functions that include pain modulation (*Swanson and Kuypers, 1980*; *van den Pol, 1999*).

Overall, 3D imaging and registration located tens of thousands of neurons across the neuraxis. Importantly, supraspinal neurons were mapped to distributed regions with broad correspondence to existing understanding of supraspinal connectivity. We also tested for variation in supraspinal numbers when injection sites were adjusted slightly to either lower lumbar (L4) or lower thoracic (T10). Interestingly, compared to L1 these cohorts showed no significant differences beyond a modest increase in CST and a modest decrease in GRN in T10-injected animals, highlighting the consistency of the approach (*Figure 1—figure supplement 5*). We conclude that 3D imaging and neuroanatomical registration provides a quantitative and global profile of neurons with spinal projections.

## Use case 1: Brain-wide comparison of supralumbar versus supracervical connectomes

Supraspinal populations can display topographic mapping with respect to spinal levels. For example, motor cortex is divided loosely into forelimb and hindlimb regions and in the RN lumbar-projecting neurons reside ventral/medial to cervical-projecting (*Flumerfelt and Gwyn, 1974*; *Tennant et al., 2011*; *Sahni et al., 2021*). More recently, retrograde labeling from lumbar and cervical spinal cord has also revealed segment-specific targeting in cerebellospinal and V2a-expressing brainstem spinal populations (*Sathyamurthy et al., 2020*; *Usseglio et al., 2020*). For many other supraspinal inputs, however, it is less clear whether projections predominantly target cervical levels, lumbar levels, or both. We therefore performed 3D imaging and registration in animals that received cervical injection of AAV2-retro-H2B-mSc and lumbar injection AAV-retro-H2B-mGL, enabling within-animal comparison (*Figure 2A–M*). Spatial registration in whole-brain data was not precise enough for definitive co-localization, and dual expression within single cells is considered in a separate analysis below. Here, we focused on overall differences in cell counts between mSc signal (cervical) and mGL signal (lumbar) in various brain regions. For each region, we calculated an index of lumbar projection by dividing the number of cells that were labeled by lumbar injection by the total number of supraspinal neurons in that region (lumbar plus cervical). Supraspinal regions that terminate mostly in cervical spinal cord would display a low lumbar projection index, whereas regions that predominantly target lumbar would display higher values, although with the caveat that fibers of passage could take up cervically injected AAV, which is considered below.

The brain as a whole had an average lumbar projection index of just 26.5% (±2.6% SEM), indicating a brain-wide tendency for many axons to terminate in cervical regions without continuing caudally to lumbar segments. Numerous brain regions deviated from this average, however. The medullary reticular formations (MDRNd and MDRNv, located caudally to the gigantocellular reticular nuclei) and the laterally positioned parvocellular nucleus both showed very low indexes of lumbar projection, indicating largely cervical termination. Interestingly, these regions have been recently implicated in

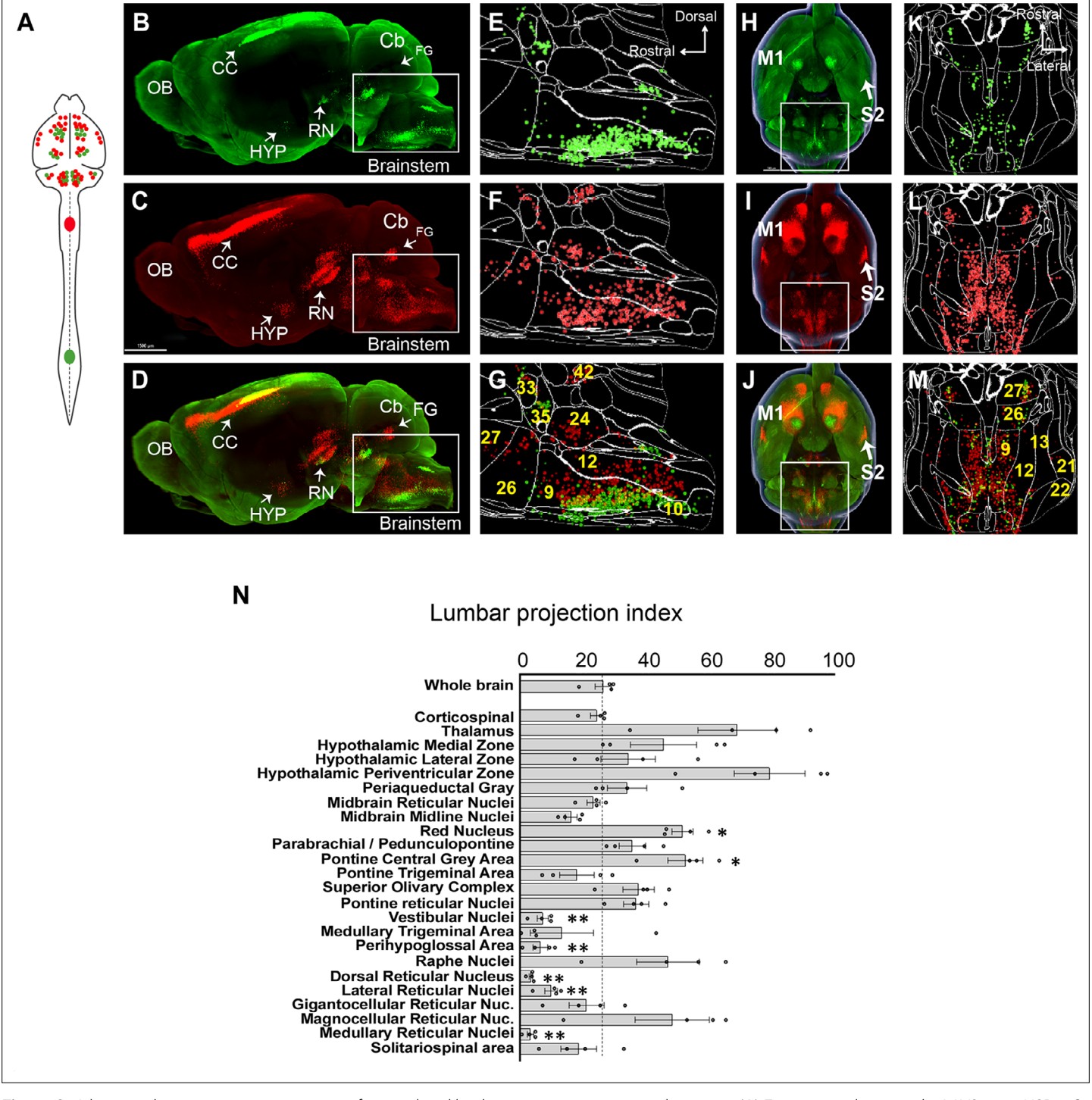

**Figure 2.** A brain-wide quantitative comparison of cervical and lumbar-projecting supraspinal neurons. (**A**) Experimental approach. AAV2-retro-H2B-mSc and mGL were delivered to C4 and L1 SC, respectively, followed 4 weeks later by brain clearing, light sheet microscopy, registration, and quantification. (**B–G**) Lateral view of brain and cellfinder output from brainstem regions. (**B, E**) show H2B-mGL (lumbar), (**C, F**) show H2B-mSc (cervical), and (**D, G**) show the overlay. Note the greater abundance of cervical signal in dorsal brainstem. (**H–M**) Horizontal brain views and cellfinder output of the same animals as (**B–G**). (**H, K**) show H2B-mGL (lumbar), (**I, L**) show H2B-mSc (cervical), and (**J, M**) show the overlay. Note the relative abundance of cervical label in more lateral brainstem. (**N**) Quantification of the lumbar projections with the number of lumbar-projecting neurons in each brain region normalized to the total number of supraspinal nuclei detected in that region. Some regions differ significantly from the overall average of the brain. **p<0.01, one-way ANOVA with post-hoc Sidak's. N = 4 animals. Scale bar (**B–D**) 1500 µm; (**H–J**) 1000 µm; (**O**) left pictures 1000 µm, middle 500 µm, right 30 µm. n =

*Figure 2 continued on next page*

*Figure 2 continued*

4 biological replicates per group. OB, olfactory bulb; CC, cerebral cortex; M1, motor area 1; HYP, hypothalamus; RN, red nucleus; Cb, cerebellum; FG, fastigial nucleus; P, pons; M. medulla. Scale bars are 1 mm, except 100 µm in 0 panel 2.

The online version of this article includes the following figure supplement(s) for figure 2:

**Figure supplement 1.** Topographic mapping of the corticospinal projection is arranged as concentric rings with caudal projections interior.

forelimb-reaching behavior, a function that would be consistent with predominantly cervical projections (*Esposito et al., 2021*; *Ruder et al., 2021*). The 3D registration approach also revealed an overall topographic pattern in the brainstem in which ventrally located populations projected to both lumbar and cervical regions, whereas more dorsally located populations rarely took up lumbar-injected label and were thus predominantly cervical (*Figure 2B–G*). The nucleus prepositus and Roller nucleus, dorsally located in the medulla, showed mostly cervical projections, possibly related to their known role in gaze tracking (*Figure 2N*; *Chiang et al., 2019*). In contrast, the pontine reticular formations showed a more balanced distribution. The RN was also relatively balanced but showed a topographic pattern in which ventral-medial neurons projected to lumbar cord (*Figure 2B–D*; *Wang et al., 2018*; *Flumerfelt and Gwyn, 1974*; *Liang et al., 2021*). Neurons near the pontine central gray, including Barrington's nucleus, showed relative enrichment for lumbar labeling, consistent with known innervation of lumbar circuitry (*Figure 2E–G*; *Verstegen et al., 2017*). Similarly, the paraventricular region of the hypothalamus was enriched for lumbar label, consistent with its known innervation of the IML cell column. In summary, these data are broadly supported by current understanding of supraspinal topography while providing region-by-region indexes of lumbar targeting for diverse supraspinal areas.

The analyses above reveal regional patterns of projection but lack the spatial resolution to co-localize signal in individual cells. In brain regions that project axons to both cervical and lumbar spinal cord, a key question is the extent to which individual cells innervate each level separately as opposed to dually innervating both. We therefore adopted an alternative confocal-based approach to achieve the needed cellular resolution. As previously, supraspinal neurons were retrogradely labeled by lumbar AAV2-Retro-H2B-mGL and cervical AAV2-Retro-H2B-mSc (*Figure 3A*). Control animals received injection of both fluorophores to lumbar spinal cord. Three weeks later, mice were perfused and the brain was sectioned into 1.5 mm slices in the transverse plane, which allowed each section to be imaged in its entirety by confocal microscopy at high resolution and without chromatic aberrations, thus allowing precise cell-by-cell assessment of retrograde label (*Figure 3B and C*). Because the slice-based imaging was incompatible with the whole-brain registration pipeline, supraspinal populations that included lumbar-projecting neurons were identified manually, as illustrated in *Figure 3—figure supplement 1*. For each of the 11 supraspinal brain regions, the number of single- and dual-labeled nuclei was quantified (see *Figure 3—figure supplement 2* illustrating region selection and quantification).

We focused our analysis on the degree to which neurons that innervated lumbar spinal cord, identified by mGL signal, also took up cervically injected mSc, indicating dual innervation. Three control animals received lumbar injection of titer-matched AAVs expressing H2B-mGL and H2B-mSc. In these, 65.8% (±0.9% SEM) of mGL-labeled neurons co-expressed mSc, establishing a baseline expectation for co-detection (*Figure 3—figure supplement 2A–G*). The finding that not all mGL-labeled nuclei co-expressed mSc even when injected to the same location likely reflects the more effective labeling by mGL noted previously (*Figure 1—figure supplement 2*). In experimental animals, we detected an average of 25,748 (±1599 SEM) lumbar-projecting neurons across all eleven brain regions, and of these an average of 41.7% (±1.3% SEM) co-expressed mSc. There was substantial variability between different brain regions, however. For example, consistent with a recent report (*Sathyamurthy et al., 2020*), in the DCN only 8.5% (±1.0% SEM) of lumbar-projecting cells co-expressed cervical mScarlet. Similarly, only 14.4% (±2.5% SEM) of lumbar-projecting rubrospinal neurons expressed mSc, consistent with established patterns of mostly segregated topography (*Figure 3D and E*). Importantly, the overall separation of mSc and mGL signal in these regions indicates that fibers of passage in cervical spinal cord mostly failed to take up Retro-AAV2, supporting prior conclusions by our lab and others that AAV2-retro is primarily taken up by collaterals or synaptic terminals (*Wang et al., 2018*; *Steward et al., 2021*; *Sathyamurthy et al., 2020*). In striking contrast, other brain regions displayed rates of mSc signal in lumbar-labeled neurons that approached the levels we measured with direct co-injection, for example, the LHA (54.8% [±4.3% SEM]) and gigantocellular reticular formation (48.5%

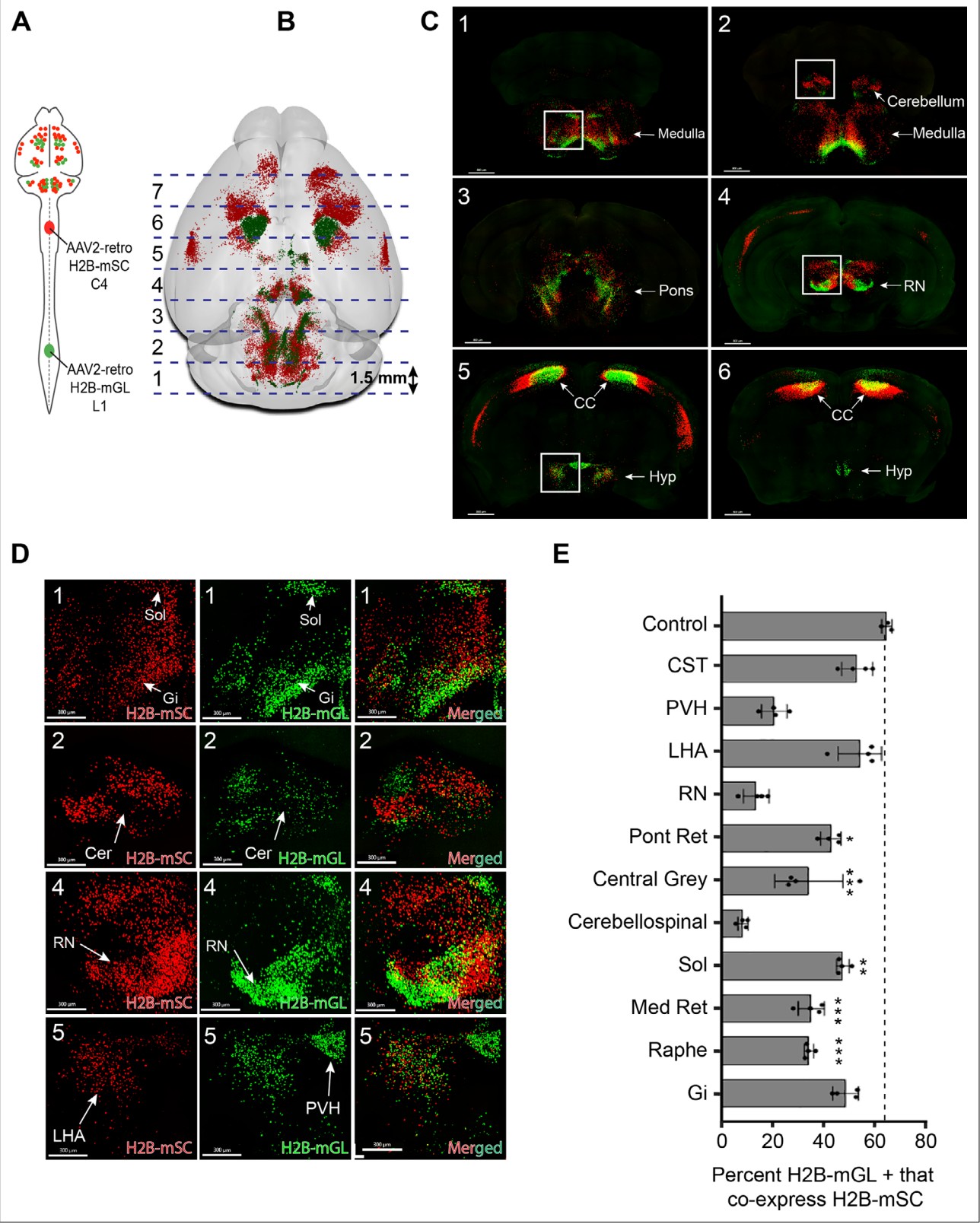

**Figure 3.** 3D confocal microscopy reveals rates of dual cervical/lumbar targeting in various supraspinal regions. (**A, B**) Experimental approach. AAV2-retro-H2B-mSc and mGL were delivered to C4 and L1 SC, respectively, followed 3 weeks later by preparation of 1.5 mm sections, tissue clearing, imaging by spinning disk confocal microscopy, spot detection, and quantification. Dotted lines show the approximate locations of slice preparation. (**C**) Coronal views of the corresponding sections imaged by confocal microscopy. Insets indicate higher magnifications in (**D**). (**D**) Higher magnification of

*Figure 3 continued on next page*

*Figure 3 continued*

specific regions from sections 1, 2, 4, and 5. Left panels show H2B-mSC-positive nuclei, middle panels show H2B-mGL nuclei, and the right panel shows merged images. (**E**) Quantification of the percent of lumbar projecting (H2B-mGL+) nuclei that co-express H2B-mSc in selected brain regions compared to control animals that receive co-injection of H2B-mSc and H2B-mGL to lumbar spinal cord. OB, olfactory bulb; CC, cerebral cortex; M1, motor area 1; HYP, hypothalamus; RN, red nucleus; Cb, cerebellum; Sol, solitary nucleus; Pons, pons; LHA, lateral hypothalamus; PVH, paraventricular hypothalamus. **** $p<0.0001$, one-way ANOVA with post-hoc Sidak's. N = 4 biological replicates per group. Scale bars (**C**) 800 μm; (**D**) 300 μm.

The online version of this article includes the following video and figure supplement(s) for figure 3:

**Figure supplement 1.** Selection of supralumbar regions for quantification of dual lumbar/cervical innervation.

**Figure supplement 2.** Quantification of retrograde co-expression of AAV2-Retro injected to the same spinal level.

**Figure supplement 3.** Retrograde signal from cervical spinal cord is dimmer in cells that are co-labeled from lumbar spinal cord.

**Figure 3—video 1.** A video illustrating confocal whole-slice imaging, selection of supraspinal regions of interest, and detection of single- and double-labeled cell nuclei.

https://elifesciences.org/articles/76254/figures#fig3video1

**Figure 3—video 2.** A video illustrating relatively dim retrograde label from cervical spinal cord in lumbar-projecting corticospinal neurons.

https://elifesciences.org/articles/76254/figures#fig3video2

[±2.5% SEM]) (*Figure 3D and E*). Most brain regions showed intermediate rates of co-labeling that differed significantly from dual-injected control. In summary, this approach reveals a diversity of innervation patterns by lumbar-projecting neurons, ranging from nearly exclusive lumbar innervation to predominantly dual lumbar/cervical innervation.

In the cortex, as expected, lumbar-projecting neurons were absent from the lateral S2 and the more rostral forelimb area (RFA) and instead were concentrated in a subregion of M1, centered medial and caudal to the main mass of cervically labeled neurons (*Steward et al., 2021*; *Sahni et al., 2021*). Interestingly, a rim of cervically labeled cells completely surrounded the lumbar region (*Figure 2O*, *Figure 2—figure supplement 1*). Thus, lumbar-projecting CST neurons can be described most accurately not as a separate caudal or medial population, but rather as nested within a broader region of cervical-projecting neurons (*Steward et al., 2021*; *Sahni et al., 2021* ). Importantly, this region also contained mSc-labeled neurons, and of the neurons that projected to lumbar cord 53.0% (±3.0% SEM) were also labeled from cervical, indicating dual projection (*Figure 3E*). This degree of co-labeling in CST neurons was initially surprising and at odds with visual impressions from light sheet images produced by our lab and others, which suggested that lumbar-projecting neurons mostly did not take up cervically-injected AAV2-retro (*Wang et al., 2018*; *Steward et al., 2021* ). We noticed, however, that the cervically derived label was often very dim in dual-labeled cells. Indeed, quantification confirmed that compared to neurons that project exclusively to cervical cord, neurons detected as dual-projecting showed significantly lower intensity of cervical mSc (*Figure 3—figure supplement 3A–E*). It is likely that this dim signal is detected by the current confocal-based methods but was missed in prior light sheet fluorescence microscopy (LSFM). A video illustrates this phenomenon by adjusting the brightness of mSc signal, resulting in shifting impressions of co-localization (*Figure 3—video 2*). We considered whether the dimness of cervically derived mSc in lumbar-projecting neurons could result from viral cross-interference. Unlike the mSc signal, however, the mGL signal averaged higher in dual-labeled neurons than in mGL-only cells (*Figure 3—figure supplement 3E and F*). Moreover, dual-labeled neurons from co-injected control animals showed higher average intensity than single-labeled neurons and a positive correlation in the brightness of the two fluorophores (*Figure 3—figure supplement 2H and I*). Thus, cross-suppression of expression is not a general feature of co-delivered AAV2-retro. As discussed more fully later (see 'Discussion'), one possible explanation is that viral uptake is proportional to synapse density, and the dim cervical label in lumbar-projecting neurons reflects collateralization in cervical cord that is present but sparse. On balance, the present data identify a nested population of CST neurons that projects to the lower spinal cord, many of which display a pattern of targeting that strongly favors lumbar over cervical but that is not necessarily completely exclusive (*Steward et al., 2021*, *Sahni et al., 2021*).

## Use case 2: Application to spinal injury

We next applied whole-brain imaging and quantification to questions related to SCI. First, given the large existing patient population and the practical difficulties in delivering therapeutics immediately

after injuries, a critical question for the eventual clinical use of gene therapy vectors is their capacity for effective transduction when applied in conditions of chronic injury. Supraspinal neurons can atrophy after injury (*Kwon et al., 2002*; *Chen et al., 2017*), which conceivably could interfere with the uptake or transport of virus. Conversely, injury can trigger spontaneous axonal sprouting above the site of injury (*Filli et al., 2021*; *Bareyre et al., 2004* ), which could potentiate viral uptake. Our prior work showed AAV2-retro's efficacy in some cell types immediately after injury (*Wang et al., 2018*), but this conclusion was only qualitative, applied to limited supraspinal populations, and did not examine more extended and clinically relevant time points. Prior work has shown that AAV2-retro can transduce selected brainstem populations when injected to the chronically injured spinal cord (*Asboth et al., 2021*; *Engmann et al., 2020*). To extend these prior findings across the supraspinal connectome after chronic injury, we injected AAV2-retro-H2B-mGL 6 weeks after a complete crush of thoracic spinal cord to a location 1 mm rostral to the injury (*Figure 4A*). After sacrifice, examination of the crush site in sections of spinal cord confirmed injury completeness, as evidenced by a lack of astrocytic bridges and lack of retrograde label distal to the injury (*Figure 4B*). In cleared brains, examination of retrograde mGL showed a broad distribution of signal (*Figure 4C–G*), and in all regions the nuclei counts did not differ significantly from those found previously in uninjured animals with similar thoracic injections (*Figure 4H*). These data quantitatively verify AAV2-retro's ability to effectively deliver transgenes to a wide diversity of cell types in the chronic phase of injury, meeting an important perquisite for eventual translation.

We next applied whole-brain quantification to a central challenge in SCI research, the issue of injury variability. In both the clinic and the laboratory, spinal injuries are often incomplete and leave inconsistent numbers of spared connections in each individual (*Fouad et al., 2021* ). To generate a range of injury severities, adult mice received crush injury to T10 spinal cord using forceps fitted with stoppers of defined thickness. The crush model was selected based on resource availability, prior use in assessing axon regeneration in mouse models (*Inman and Steward, 2003*; *Liu et al., 2010*; *Leibinger et al., 2021*; *Brommer et al., 2021*; *Du et al., 2015*), and its demonstrated ability to produce graded injuries by varying the stopper width (*Plemel et al., 2008*). The animal received severe injuries (0.15 mm stoppers), moderate injury (0.4 mm stoppers), and mild injury (spinal cord displaced by the forceps width but not squeezed) (*Cho et al., 2010* ). AAV2-retro-H2B-mGL was injected to L4 spinal cord 7 weeks post-injury and tissue was analyzed 2 weeks post-injection (*Figure 5A*). In selecting the L4 location, we considered that short-distance sprouting, but not long-distance axon extension, occurs spontaneously after injury (*Filli et al., 2021*; *Bareyre et al., 2004* ). Accordingly, the L4 injection site was chosen to target spared axons, while minimizing the potential for injected virus to spread above the injury location.

First, viral targeting and injury severity were assessed in spinal tissue sections (*Figure 5B–G*); images of all spinal cords are provided in *Figure 5—figure supplement 1*. Crush injuries in mice produce fibrotic scars that can be recognized by high GFAP signal surrounding a GFAP-negative core, indicating interruption of the initial astrocytic continuity (*Liu et al., 2010*; *Soderblom et al., 2013* ). We assessed the relative injury severity by quantifying the width of GFAP bridges that spanned the injury, normalized to total width of the spinal cord. As expected, mild injuries displayed elevated GFAP at the crush site but overall astrocytic continuity (92.5% ± 2.0 SEM) (*Figure 5B and C*, *Figure 5—figure supplement 2*). Severe injuries completely abolished astrocytic continuity in two animals and reduced the third to 0.73% of initial. Moderate injury produced a significant reduction in astrocytic bridging compared to mild injury (32.2%±7.0 SEM, p=0.005 ANOVA with post-hoc Sidak's) but with considerable variability (range 8.4–75.4%). Overall, as intended, the injuries displayed a wide range of severity.

To assess residual brain–spinal cord connectivity, supraspinal neurons were registered and quantified by the pipeline described above (*Figure 5H–N*). Note that although we detected some inter-animal variability in the injection location (e.g., more caudal in animals 179 and 187), our prior findings indicate this will have minimal impact on retrograde cell counts in the brain (*Figure 1—figure supplement 5*). Raw values for all brain regions are provided in *Source data 1*, and *Figure 5N* shows values normalized to region counts in uninjured mice, thus creating an index of sparing for each region. Mice that received severe injuries showed a maximum of 29 labeled cells brain-wide, confirming disruption of descending axon tracts (*Figure 5J, M and N*). In contrast, mild injuries averaged only a 43% reduction in retrograde label, with high variability between different supraspinal populations (*Figure 5H, K and N*). For example, the CST was strongly affected, averaging less than 20% sparing, while neurons

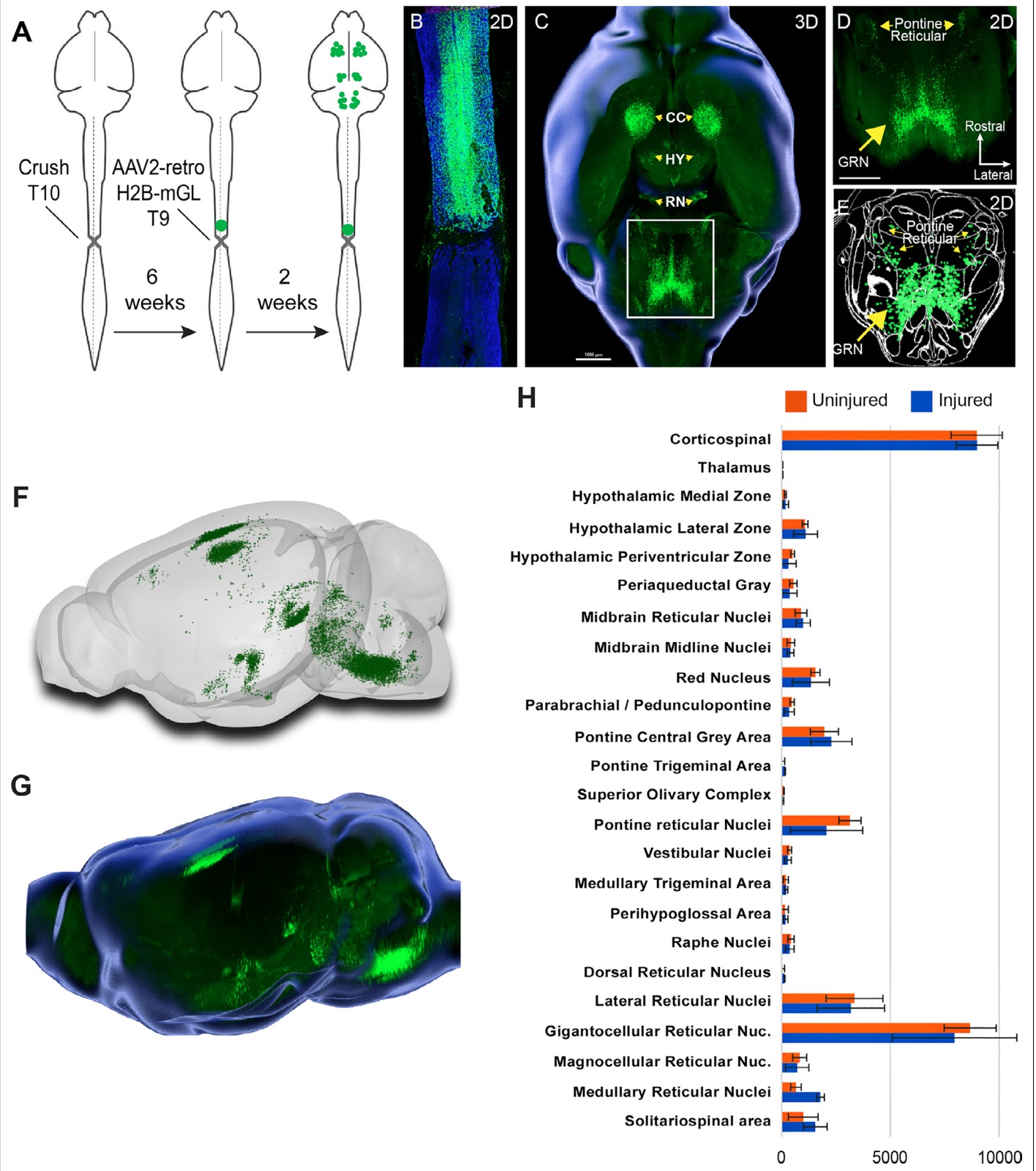

**Figure 4.** Retro-AAV2 effectively transduces neurons when delivered to the chronically injured spinal cord. (**A**) Experimental design. A complete crush injury was delivered to lower thoracic spinal cord, followed 6 weeks later by injection of AAV2-retro-H2B-mGL rostral to the injury. Two weeks post-injection, animals were euthanized, brains were cleared and imaged with light sheet microscopy, and images were processed for registration and quantification by cellfinder and brainreg. (**B**) Horizontal spinal section stained for GFAP (blue), confirming the complete crush and verifying transduction

*Figure 4 continued on next page*

*Figure 4 continued*

(green) rostral but not caudal to the injury. (**C–E**) Horizontal views of the brain from the same animal of section in (**B**). (**C**) shows a 3D overview, (**D**) shows a higher magnification view of the brainstem, and (**E**) shows cellfinder output with retrograde mGL detection in green. (**F**) Brainrender output showing lateral views of the same brain in (**C**). (**G**) Imaris 3D-generated equivalent lateral view of the brain in (**F**). (**H**) Quantification of retrograde nuclei detected in 25 brain regions, comparing uninjured animals (orange) to chronically injured animals (green). No regions displayed statistical differences (p>0.05, two-way ANOVA with post-hoc Sidak's). N = 4 animals per group. Scale bars are 1 mm.

near the pontine central gray and the RN averaged 83.3 and 75.1% sparing, respectively (*Figure 5N*). The moderate injury group showed a brain-wide average of 24.2% sparing, also with high variability between animals (range 3.4–36.3%) (*Figure 5I, L and N*). As in the mildly injured animals, CST neurons were affected more strongly than other populations such as the PGC and RN, although the animal with the highest overall sparing showed an unusual pattern of nearly 70% persistence of the cortico-spinal tract (*Figure 5N*). In summary, these data confirm the ability of brain-wide analysis to detect overall sparing differences in groups of animals that received injuries of different severity, and more importantly to detail differences in the injuries' effects on individual animals and on individual cell populations.

We next asked how indexes of sparing correlate with functional recovery from spinal injury as assessed by the Basso Mouse Scale (BMS) motor score, a well-established measurement of hindlimb function and interlimb coordination (*Basso et al., 2006*). The BMS is a nonlinear, progressive scale between 0 (no hindlimb movements) and 9 (normal locomotion). Importantly, scores <4 indicate an absence of weight-bearing steps from a paw placed in the normal plantar orientation, while scores of ≥4 indicate locomotion with gains in frequency, paw placement, and limb coordination. As expected, BMS scores averaged lower in animals that received severe injury versus moderate or mild (*Figure 6— figure supplement 1*). Notably, however, variation in BMS scores was evident within severity groups.

We hypothesized that variability in motor recovery may reflect in part inter-animal differences in the amount of sparing of supraspinal neurons. Of the 12 animals, 4 recovered to a level of consistent plantar stepping (BMS ≥ 5), whereas 8 animals displayed no or infrequent stepping (BMS ≤ 3.5). We reasoned that comparing spared neurons between the two groups could provide a first-pass indication of the involvement of different brain regions. Specifically, instances of high sparing in impaired animals would indicate that a region is insufficient to confer recovery of stepping, while instances of near-ablation in animals with high performance would indicate that the region is dispensable. Thus, one signature of a stepping-relevant region would be nonoverlapping values of sparing between impaired and high-performing animals. We therefore visualized the range of values in each brain region by plotting the number of spared neurons against the final BMS score, followed by testing for significant differences in sparing between impaired and high-performing groups using thresholds adjusted for multiple comparisons (*Figure 6A–C*).

In most brain regions, the number of spared neurons did not differ significantly between impaired and high-performing groups, accompanied by examples of overlap discussed above. For example, sparing in hypothalamic-spinal neurons did not differ between groups and multiple instances existed of both high-sparing/low-performance and low-sparing/high-performance (*Figure 6B*). Thus, as expected from its predominantly autonomic role, the amount of sparing in hypothalamic-spinal regions likely did not contribute to differences in motor recovery. Similarly, two animals that achieved high walking scores showed near-total ablation of the CST, indicating a dispensable contribution to flat-ground locomotion. Conversely, two animals showed high levels of rubrospinal sparing but almost no plantar steps (BMS 3), indicating that substantial rubrospinal sparing was insufficient to mediate recovery of stepping motions. In contrast, sparing in supraspinal neurons located in the dorsal pontine region, most notably the pedunculopontine region, differed significantly between the two groups, with the most damaged animal in the high-performing group exceeding the maximum value in the impaired group (*Figure 6B*). Interestingly, the number of spared neurons in the cervical spinal cord also differed significantly and showed the widest gap between the maximum impaired value and the minimum high-performing value, thus highlighting the potential importance of cervical neurons as a supralumbar control center (*Zholudeva et al., 2021*). Images of spared propriospinal neurons in cervical spinal cord are provided in *Figure 6—figure supplement 2*. Overall, these data highlight brain regions with a cross-animal pattern of sparing that is consistent with a role in functional recovery,

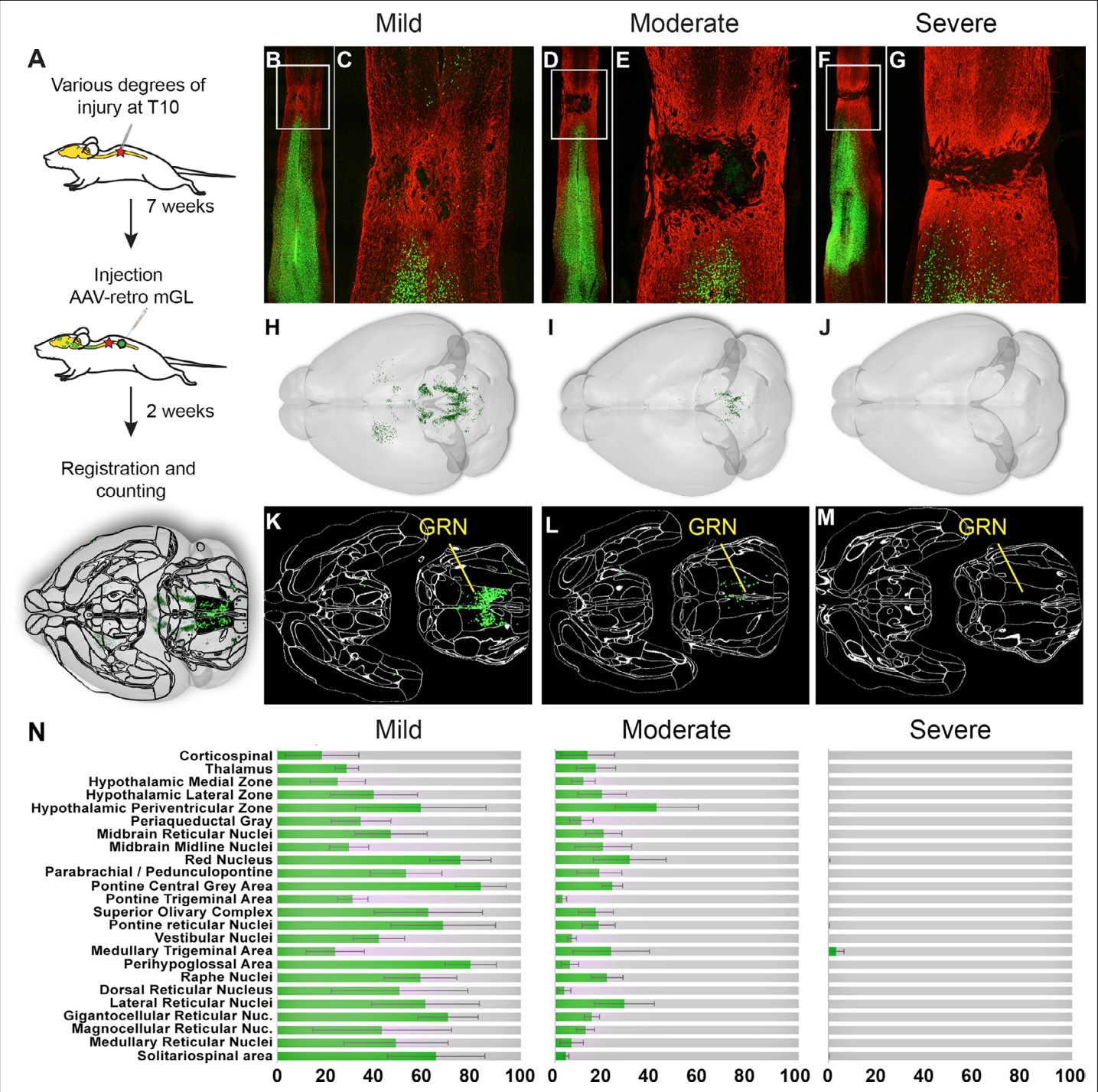

**Figure 5.** Brain clearing and registration quantifies region-specific sparing across a range of spinal injury severities. (**A–G**) Animals received thoracic crush injuries of controlled widths, followed 8 weeks later by injection of AAV2-retro-H2B-mGL to L4 spinal cord. Horizontal spinal cord sections show mild (**B, C**), moderate (**D, E**), or severe (**F, G**) injury, with GFAP in red and viral transduction in green. (**H–M**) Horizontal views of the brains from the same animals of the spinal cord sections above, showing progressive reduction in the number of retrogradely labeled neurons as injury severity increases. (**H–J**) show brainrender depictions of whole brain, (**K–M**) show one 2D cellfinder plane output with brain regions outlined and detected mGL in green on the right panels. (**N**) Quantification of percent sparing in identified brain regions, with the average value from uninjured animals set as 100. N = 3 mild, 6 moderate, and 3 severely injured animals. GRN, gigantocellular reticular nucleus. Scale bars are 1 mm.

The online version of this article includes the following figure supplement(s) for figure 5:

**Figure supplement 1.** Horizontal spinal sections with GFAP labeling for all injured animals.

**Figure supplement 2.** Graded spinal injuries reduce astrocytic continuity across the injury site.

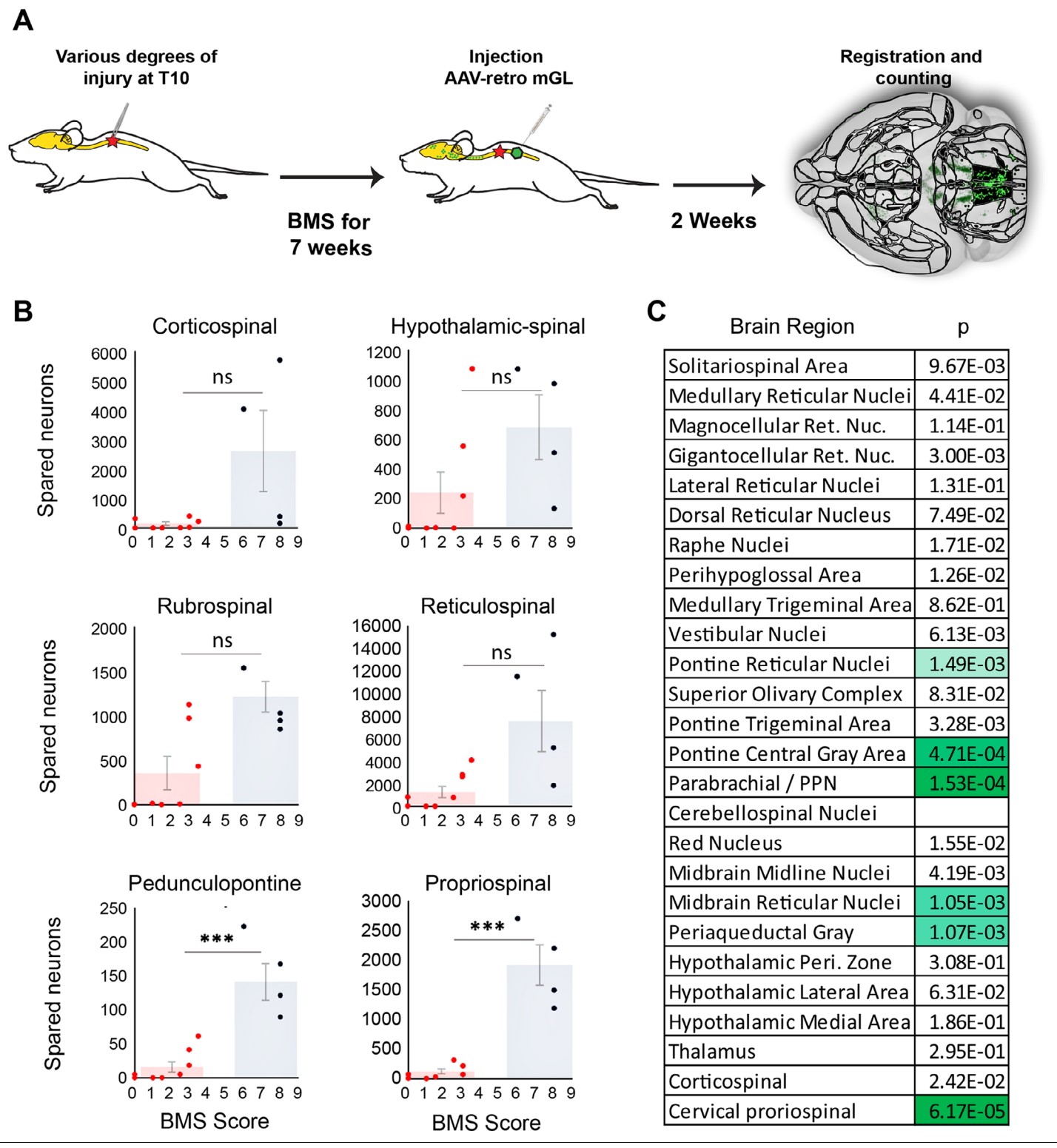

**Figure 6.** Mice that recover frequent plantar placement after injury differ significantly in neuronal sparing in selected brain regions. (**A**) Experimental design. Adult mice received graded spinal cord injuries followed by weekly Basso Mouse Scale (BMS) testing. Then, 7 weeks post-injury animals received lumbar (L4) injection of retro-AAV2-H2B-mGL, followed 2 weeks later by perfusion, 3D imaging of brains, registration, and region-specific quantification of spared neurons. (**B**) Mice were divided into two groups according to whether they regained frequent plantar stepping motions (blue, BMS > 5) versus no or infrequent plantar steps (red, BMS < 4) and then tested for significant differences in spared neurons in each of the 26 brain and spinal regions. Example scatterplots with overlaid averages show overlap in spared counts in most brain regions, but group separation and significant

*Figure 6 continued on next page*

*Figure 6 continued*

differences in selected regions including the pedunculopontine nucleus (PPN) and cervical spinal cord. (**C**) The significance level of sparing differences between the two groups across all brain regions. N = 4 skilled steppers, 8 impaired. p determined by paired *t*-tests with significance threshold set by Bonferroni's correction for multiple comparisons.

The online version of this article includes the following figure supplement(s) for figure 6:

**Figure supplement 1.** Spinal injuries of different severities result in different levels of locomotor recovery.

**Figure supplement 2.** The number of spared propriospinal neurons in cervical spinal cord correlates with locomotor recovery after spinal injury.

providing a preliminary indication of the potential utility of whole-brain imaging and quantification to partially explain variability in functional outcomes after spinal injury.

## Discussion

We have assembled an experimental pipeline and associated web-based resource that provides comprehensive quantification and visualization of neurons that project from the murine brain to specific levels of the spinal cord. Supraspinal-projecting populations are numerous and diverse, yet prior publications, particularly in the SCI field, have focused disproportionately on specific sets of nuclei and a handful of major pathways (*Blackmore et al., 2021*). The approach presented here is needed to spur progress by clarifying the complex arrangement of supraspinal neurons and providing a roadmap for a practical means to assess post-injury connectomes across the entire brain in numerous animals within an experimental study. This approach opens the door to more comprehensive analyses of changes in supraspinal connectivity in response to disease and injury, and conversely to profile without bias the brain-wide efficacy of pro-regenerative therapeutics.

### Improved cell detection in optically cleared tissue

An essential element of this approach is the deployment of newer-generation FPs, mScarlet, and mGreenLantern, with nuclear targeting (*Bindels et al., 2017*; *Campbell et al., 2020* ). Compared to prior vectors, we showed strongly enhanced detection of retrogradely transduced neurons, most notably in the brainstem, an important source of supraspinal control. This likely reflects the relative concentration of fluorescence in the nucleus, augmented by slower protein turnover and reduced interference from intervening axon tracts. It is likely that prior work, including our own and a recent description of CST neurons in cleared brains, underestimated the number of retrogradely transduced neurons (*Wang et al., 2018*; *Steward et al., 2021* ). Thus, nuclear-localized mSc and mGL expand the tool kit for neuronal labeling in cleared tissue and allow flexibility in dual-labeling experiments.

### Toward a more accessible supraspinal connectome

We designed our workflow to meet two central objectives: quickly assign supraspinal neurons to precise locations and disseminate this information in a cohesive, practical fashion (*Tyson and Margrie, 2021*). Prior work has employed retrograde tracing and manual scoring of brain sections to meticulously catalog supraspinal populations, and in most cases this foundational work should be considered definitive (*Leong et al., 1984*, *Liang et al., 1997*). Indeed, the broad concordance of these prior human-curated efforts with our automated registration approach, highlighted in 'Results,' provides essential validation (*Nudo and Masterton, 1988*; *Kuypers and Martin, 1982*; *Leong et al., 1984*; *Liang et al., 1997*; *Flumerfelt and Gwyn, 1974*; *Ruder et al., 2021*; *McCrea and Horn, 2006* ). As one standout example, the detection of cervically projecting neurons in the amygdala was initially surprising, but a literature search revealed a decades-old description of this pathway in monkeys and more recently in mice (*Liang et al., 1997*; *Mizuno et al., 1985*). Indeed, prior description can be found for nearly all populations detected in our workflow. The disparate and often nonquantitative nature of prior work, however, presents significant challenges to effective utilization. These challenges are compounded by the anatomical complexity and the difficulty of synthesizing anatomical data presented in 2D or described in reference to anatomical landmarks. The pipeline presented here marks an important step in overcoming these challenges by offering standardized detection, registration, and quantification of supraspinal populations across the brain. In addition, the ability to visualize populations in 3D lowers conceptual barriers by generating intuitive insights between the data and

brain regions. By focusing on connectivity between the brain and spinal cord, this novel resource fills a gap in existing web-based neuroanatomical atlases, which focus mostly on intra-brain circuitry. Illustrating the utility, next we highlight several key insights of high significance to both basic neuroanatomy and to preclinical research that were enabled by the new approach.

## Global assessment of cervical/lumbar topographic mapping from supraspinal neurons

As noted previously by us and others, the present data indicate that AAV2-retro is taken up by axon terminals but minimally or not at all by fibers of passage (*Wang et al., 2018*; *Steward et al., 2021*; *Sathyamurthy et al., 2020*). For example, injection of AAV2-retro to cervical spinal largely fails to label cerebellospinal and rubrospinal neurons that innervate lumbar regions. Although this may partly reflect the gray matter injection coordinates, it should be noted that virus readily diffused to contralateral spinal cord, as evidenced by symmetrical retrograde label in the brain, and therefore likely also spread to ipsilateral fibers of passage. Thus, limited uptake by axons of passage likely reflects a property of AAV-retro, which we speculate may be a synaptic mechanism of uptake.

Whatever the mechanism, this property of AAV2-retro affords an opportunity to classify supraspinal neurons based on innervation of cervical gray matter, lumbar gray matter, or both. This distinction is an important parameter of neural function that sets a baseline prediction for the capacity to coordinately regulate spinal multiple levels or conversely to achieve level-specific communication. Classical anatomical studies have shown extensive collateralization in several descending tracks with some interspecies differences (*Honsay, 1952 Hayes and Rustioni, 1981*; *Martin et al., 1981*; *Huisman et al., 1982*; *Huisman et al., 1981*). Consistent with prior work, we find that several supraspinal neuronal populations project to both the cervical and lumbar cord and the distribution of double-labeled neurons varies across nuclei (*Sathyamurthy et al., 2020*; *Usseglio et al., 2020*; *Huisman et al., 1982*; *Huisman et al., 1981* ). Some of these populations, such as the CST, show a higher innervation density in cervical compared to lumbar cord (*Sahni et al., 2021*; *Fiederling et al., 2021*). Among those neurons that extend axons to lumbar spinal cord, we detected cervically derived label in about 42%. When adjusted for the imbalance in labeling efficacy between fluorophores (mSc co-labeled only about 65% of the mGL-labeled cells even when co-injected), this finding indicates that across the whole brain a slight majority of lumbar-projecting supraspinal axons likely also collateralize to some extent in cervical spinal cord. This overall conclusion, however, must be qualified in two important ways, however. First, the degree of dual labeling varied widely between populations. For example, we detected cervical label in less than 15% of supralumbar neurons in the deep cerebellar and red nuclei compared to more than 50% in the lateral hypothalamus and gigantocellular reticular. By using a consistent methodology, our approach has revealed large differences in the relative tendency of different supralumbar populations to collateralize in cervical gray matter.

The second qualifier regards the wide range of cervically derived signal intensity within lumbar-projecting neurons. In the CST, lumbar-projecting neurons often took up cervical AAV2-retro injected to cervical spinal cord, indicating some dual innervation, yet the resulting fluorescence was generally quite dim. An explanation may be that viral uptake is proportional to the degree of collateralization, such that neurons with very few collateral branches near the injection site receive a lower copy number. We favor this model over alternatives, such as cross-interference between viruses, because (1) when two viruses are co-injected the fluorescence intensity of each correlates positively, not negatively, with the other and (2) the lumbar-derived signal, unlike the cervical, is no dimmer in dual-labeled CST neurons than lumbar-only neurons. The model that retrograde labeling is proportional to the amount of innervation would predict that lumbar-projecting CST axons may send collaterals to cervical targets but at a density that is lower than in the lumbar cord, and also lower than CST neurons that project only to cervical cord. Indeed, this is the pattern reported for CST neurons by anterograde tracing in two recent studies (*Steward et al., 2021*; *Sahni et al., 2021*). More broadly, this model would emphasize that dual innervation of spinal regions is not a binary distinction but rather should be understood as a continuum, a graded tendency of lumbar-bound axons to collateralize by varying amounts in cervical tissue. This model also points toward an important technical implication: conclusions about rates of dual innervation will necessarily depend on the threshold of detection. Indeed, based on the observed range of intensities in CST neurons, we can estimate that in a hypothetical scenario in which only the brightest 20% of neurons labeled from cervical cord were detected,

the estimated rate of dual innervation would decrease by nearly 80% as dual-labeled neurons with dim cervical label disproportionally 'drop out.' We speculate that this phenomenon may help explain why our current estimates of dual innervation by lumbar-projecting neurons trend higher than prior reports: we found 8% in the cerebellospinal versus complete segregation previously (*Sathyamurthy et al., 2020*), over 50% in the CST compared to almost none previously (*Steward et al., 2021*), and more than 40% in the gigantocellular compared to about 20% previously (*Usseglio et al., 2020*). One explanation may be that with bright, nuclear-localized fluorophores and 3D confocal microscopy we detect a larger proportion of lumbar-projecting neurons with only dim labeling from cervical targets. Indeed, our own prior assessment of dual innervation by CST neurons based on less sensitive light sheet microscopy likely missed these dimmer cells. In summary, our data suggest a working model in which some supraspinal neurons that innervate the lumbar spinal cord also strongly innervate cervical targets while many others show strong preferential innervation of lumbar targets, but with nonzero levels of cervical collateralization and residual uptake of AAV2-retro.

## Retrograde gene delivery to chronically injured supraspinal neurons

Our findings have important implications for research that aims to address the needs of individuals that suffer from supraspinal disruptions such as SCI. Besides the well-studied locomotor and fine motor deficits, SCI also affects sensation, bladder and bowel control, sexual function, basic postural control, cardiovascular tone, thermoregulation, and even metabolism (*Anderson, 2004* ). Supraspinal populations that serve these functions are known, yet their response to injury and to attempted pro-regenerative strategies is largely uncharacterized (but see *Adler et al., 2017* for an example of CNS-wide profiling). Treatments can advance even to clinical trials with limited information on how or if they influence axon growth in tracts beyond the major motor pathways. Thus, although much progress in this direction has been made, there arguably remains a mismatch between the varied concerns of individuals suffering from SCI and the narrower anatomical focus of SCI research. In this context, the brain-wide workflow presented here offers an example of a practical means to expand the study of nonmotor systems, and to populations that likely modulate the major motor pathways. Specifi-cally, as pro-regenerative treatments are tested in preclinical mouse models, the whole-brain pipeline presented allows the response of diverse cell types to be monitored with the throughput needed for SCI studies. As a first illustration of this strategy, we used the pipeline to determine how the chronic injury state impacts transduction by retrograde vectors. This is an important consideration for the translational prospects of gene therapy approaches to treat CNS damage. We showed quantitatively that gene delivery remained highly effective and widespread in chronic spinal injury, information that is needed to proceed with gene therapy-based treatments of the broad concerns discussed above. In summary, adoption of this connectome-level approach has the potential to sharpen the predictive power of preclinical work and help to better align it with the concerns of individuals with SCI.

## Implications for the neuroanatomical–functional paradox in spinal injury research

Another central challenge in the SCI field is the neuroanatomical–functional paradox, which refers to the fact that the size of lesions in the spinal cord is poorly predictive of functional outcomes (*Fouad et al., 2021* ). This unpredictability of SCI outcomes is a major stumbling block that has likely contrib-uted to challenges of reproducibility in the field (*Steward et al., 2012*). The recovery of various func-tions after spinal injury is almost certainly impacted by the amount of residual supraspinal connectivity from specific brain regions, yet the field has lacked a practical means to quantify this key variable from most supraspinal populations. We now demonstrate comprehensive quantification of the variability of residual brain–spinal connectivity across injury types and between animals. Prior work has shown that detailed analysis of the spinal injury site, for example, the amount of spared white matter, can partially explain variation in functional recovery (*Fouad et al., 2021 Loy et al., 2002*; *Schucht et al., 2002* ). We speculate that complementing these existing approaches with complete quantification of residual supraspinal connectivity could significantly boost the ability to predict and explain differen-tial outcomes. It is important to note that even a consistent correspondence between a population's sparing index and the degree of functional recovery does not necessarily imply functional involvement; it could, for example, be caused by axon trajectories in proximity to more functionally relevant tracts. In cases where the absolute number of supraspinal axons is low, or where prior findings contradict

functional involvement, the correlations are likely by-products. The value of associating functional recovery with residual connectivity across the brain is to rapidly generate and prioritize functional hypotheses, which must be synthesized with existing information. Ultimately the functional contribution of discrete populations must be tested directly, for example, with targeted chemogenetic or optogenetic inhibition. The value of the current approach is to prioritize those efforts by directing attention toward neural populations whose pattern of sparing is most consistent with a functional role.

As one example, our data detect a direct projection from the PPN and the lumbar spinal cord and a high correlation between maintenance of this pathway with locomotor function after SCI. Interestingly, the PPN lies within the MLR, an area that has been recognized for more than 50 years as an important area for locomotion (*Gatto and Goulding, 2018*; *Shik et al., 1969*) but which has received limited attention in SCI research. Thus, the correlational analysis presented here raises the hypothesis that in addition to rubrospinal and reticulospinal projection, the maintenance and regeneration of PPN-spinal projections may impact locomotor recovery from spinal injury. By directing attention toward lesser-studied but potentially significant regions, the integrated approach presented here can point toward the key functional experiments needed to help resolve the variability in outcomes that currently challenge the field.

Important caveats to the approach should be considered. First, although the crush model has been used to assess axon regeneration in the mouse (*Liu et al., 2010*; *Leibinger et al., 2021*; *Brommer et al., 2021 Du et al., 2015*), it must be acknowledged that it does not mimic the acute impact involved in most human injuries. For example, it was noted previously that spinal contusion can leave an outer rim of spared tissue, which may favor residual brainstem–spinal connectivity (*Asboth et al., 2021*). Thus, the results presented here should be understood as a proof of concept that whole-brain analyses can detect region-specific differences in sparing across injury severities, but an important future direction will be to evaluate sparing in contusion models that better mimic human injury (*Cheriyan et al., 2014*). Second, automated detection of nuclei likely remains imperfect and is impacted by the image quality of LSFM, notably stretching in the Z plane. Continued improvements with more isotropic acquisition in light sheet microscopy and in trained detection of nuclei will likely resolve these lingering issues (*Chakraborty et al., 2019*; *Strack, 2021*). A third caveat regards viral tropism, for example, AAV2-retro appears less effective at transducing serotonergic cell types, thereby limiting the assessment of raphe-spinal projections (*Wang et al., 2018*). This limitation will likely be addressed as additional retrograde variants are made (*Davidsson et al., 2019*). Fourth, it is important to note that brain injury and disease can potentially alter the size of individual brain regions, but that the process of image registration smooths and obscures these differences (*Niedworok et al., 2016*). Thus, the workflow here is not suited to detect region-specific volumetric changes, and care should be taken in interpreting the results of registration in areas where large changes may have occurred. Finally, we have applied this approach only to descending inputs to the spinal cord, and not to ascending tracts. In principle, a similar retrograde strategy could quantify neurons that give rise to ascending input, and a promising future direction would be to incorporate this information into predictive models for function after partial spinal injury. However, while these and other future developments are likely to further improve the approach, the present iteration provides information on an unprecedented scale and has yielded new insights into the complexity of supraspinal populations and their variable response to spinal injury.

# Materials and methods
## Animal information

All animal procedures were approved by the Marquette University Institutional Animal Care and Use Committee and complied with the National Institutes of Health Guide for the Care and Use of Laboratory Animals. Adult female C57BL/6 mice (6–8 weeks old, 20–22 g) were used for these experiments. Age at the day of surgery was 8 weeks and mean weight was 20 g. Groups for initial fluorophore optimization were mScarlet cytoplasmic, L1 injected, 4 weeks: 4 animals; H2B-mScarlet, L1 injected, 4 weeks: 4 animals; H2B-mScarlet, L1 injected, 2 weeks: 8 animals; H2B-mGreenLantern, L1 injected, 2 weeks: 8 animals. Group sizes for cleared and registered brains were L1 injected: 10 animals; L3/4 injected: 5 animals; T10 injected: 4 animals; cervical/lumbar co-injected: 4 animals; chronically injured: 3 animals; moderately injured: 6 animals; mildly injured: 3 animals; severely injured: 3 animals. In the

L1-injected group with 2 weeks survival, one H2B-mGL-injected animal showed likely blockage of the injection needle as evidenced by lack of local transduction at the injection site, and one H2B-mSc-injected animal suffered tissue damage during the brain dissection. These animals were excluded, leaving seven per group. The room temperature was set at 22°C (±2°C) and room humidity was set at 55% (±10%). Mice were kept in a 12 h light/dark cycle with access to food and water ad libitum. Mice were checked daily by animal caretakers.

## Plasmid construction and cloning

We used two monomeric bright FPs of similar size that encode for mGreenLantern (*Campbell et al., 2020*) and mScarlet (*Bindels et al., 2017*) and fused in frame with the core histone H2B in the amino terminus for nuclear localization of the FPs. Both fusions were synthetically constructed (GenScript, USA). The AAV mGreenLantern was constructed first by generating a synthetic cDNA optimized to the Human codon usage. The rat gene H2B/histone H2B type 1C/E/G (accession #NP_001100822) was fused in frame to mGreenLantern with a linker of eight amino acids (PPAGSPPA) between H2B and mGreenLantern. The fusion protein was cloned into pAAV-CAG-GFP (Addgene #37825) by substituting the GFP with H2B-mGreenLantern using restriction enzymes BamHI and XhoI. For the mScarlet, the human H2B-clustered histone 11 (H2BC11) (accession #NM_021058) was fused in frame without a linker and cloned into the pAAV-CAG-tdTomato (Addgene #59462) using the sites KpnI and EcoRI at the 5′ and 3′ end, respectively. Cytoplasmic mScarlet was cloned identically but with the H2B sequence omitted. rAAV2-retro-H2B-mGreenLantern was produced at the University of Miami viral core facility at the Miami Project to Cure Paralysis, titer = $1.4 \times 10^{13}$ particles/ml. Virus was concentrated and resuspended in sterile HBSS and used without further dilution. The rAAV2-retro-mScarlet and rAAV2-retro-H2B-mScarlet was made by the University of North Carolina Viral Vector Core, titer = $4.3 \times 10^{12}$ and $8.7 \times 10^{12}$ particles/ml, respectively.

## Spinal cord surgery

rAAV-retro particles (1 µl) were injected into the spinal cord with a Hamilton syringe driven by a Stoelting QSI pump (Cat# 53311) and guided by a micromanipulator (pumping rate: 0.04 µl/min). AAV viral particles were injected at C4-C5, T10, L1, L4 vertebrae, 0.35 mm lateral to the midline, and to depths of 0.6 and 0.8 mm. For the spinal cord crush injuries, adult female C57BL/6 mice (6–8 weeks old, 18–22 g) were anesthetized by ketamine/xylazine. After laminectomy of vertebra, T10-12 forceps with stoppers of 0.15 mm (narrow gap, severe injury) or 0.4 mm (moderate injury) were used to laterally compress the spinal cord for 15 s, then flipped in orientation and reapplied at the same site for an additional 15 s. To produce mild injuries, forceps were placed laterally to the cord and within the vertebral column, resulting in some tissue displacement, but not squeezed.

## BMS scoring

Starting 1 week post-injury, mice were digitally recorded while engaging in open-field locomotion. Two blinded observers evaluated each mouse using the 10-point BMS (*Basso et al., 2006*). The BMS is a nonlinear, progressive scoring system that assesses aspects of locomotion including joint movement, plantar versus dorsal paw placement, stepping, paw placement, limb coordination, and trunk support. Differences in scores between observers triggered discussion followed by consensus scoring. When scores differed between right and left hindlimb, the BMS score is reported as the average of the two. Testing was performed weekly for 7 weeks.

## Tissue clearing and imaging

After 2–4 weeks of viral expression, the animals underwent transcardial perfusion with 0.9% saline and 4% paraformaldehyde (PFA) solutions in 1× phosphate-buffered saline (PBS) (15710, Electron Microscopy Sciences, Hatfield, PA). Whole brains and spinal cords were dissected and fixed overnight in 4% PFA at 4°C and washed three times in PBS pH 7.4, followed by storage in PBS. The dura was carefully removed and brains and spinal cords were cleared using a modified version of the 3DISCO (*Wang et al., 2018*; *Soderblom et al., 2015* ). To prepare tissue sections, brains were embedded in gelatin, fixed overnight in 4% PFA, then sliced in the transverse plane at 1.5 mm intervals, proceeding rostrally from a first cut positioned at the pyramidal decussation (Leica VT1200), then stored in PBS with.02% w/v NaAzide. Whole mouse brains were incubated on a shaker at room temperature in 50%,

80%, and twice with 100% peroxide-free tetrahydrofuran (THF; Sigma-Aldrich, 401757) for 12 hr each for a total of 2 days. Tissue sections were treated similarly but with shortened incubation times, 2 hr for the 50%, 80%, and first 100%, then overnight for the second 100%. Peroxides were removed from THF by using a chromatography column filled with basic activated aluminum oxide (Sigma- Aldrich, 199443) as previously described (*Becker et al., 2012*). Samples were transferred to BABB solution (1:2 ratio of benzyl alcohol, Sigma-Aldrich, 305197; and benzyl benzoate, Sigma-Aldrich, B6630) for at least 3 hr (whole brains) or 1 hour (sections). After clearing, whole brains were imaged the same day using light sheet microscopy (Ultramicroscope, LaVision BioTec). The ultramicroscope uses a fluorescence macro zoom microscope (Olympus MVX10) with a 2× Plan Apochromat zoom objective (NA 0.5). Image analysis and 3D reconstructions were performed using Imaris v9.5 software (Bitplane, Oxford Instruments) after removing autofluorescence using the Imaris Background Subtraction function with the default filter width so that only broad intensity variations were eliminated. The 1.5 mm sections were imaged using a high-speed confocal, Andor Dragonfly 202-2540 (Oxford Instruments), based on a Leica microscope. We used a 10× Plan Apo, NA 0.45 and W.D 2.8. The lasers were a solid-state 488 nm diode laser at 150 mW and OBIS LS 561 smart OPSS laser at 100 mW. Laser power was set at 15% and exposure time for each plane was between 30 ms for mGL and 40–50 ms for mSC. The camera used was a Sona sCMOS 4.2B-6 set at ROI size (W × H) at 1024 × 1024. All images were stitched using Imaris Stitcher Vx64 9.7.2 (Oxford Instruments). After stitching, the 3D rendering was done on Imaris 9.8.2 and the individual nuclei were counted using the spot function at 4 µm diameter in XY and 8 µm in Z. The segmentation was set based on the histogram values and manually adjusted to cover all spots in the red and green channels. Background spots were manually deleted and the double-labeled nuclei automatically detected based on distance from H2B-mGL to H2B-mSC set to 4 µm.

## Imaris reconstructions

Image analysis and 3D reconstructions were performed using Imaris v9.5 software (Bitplane, Oxford Instruments) after removing autofluoresence using the Imaris Background Subtraction function with the default filter width so that only broad intensity variations were eliminated. Additionally, the entire brain was defined as an ROI in order to mask all background fluorescence outside the spinal cord surface. Artifact and nonspecific fluorescence surrounding the brain were segmented and removed using the automatic isosurface creation wizard based upon absolute intensity. Voxels contained within the created surface were set to zero, and the remaining mask was used for all further analysis. Automatic segmentation of nuclei within specified ROIs was applied using the spots detection function and later superimposed on a maximum intensity projection volume rendering of the tissue. For some of the figures, surfaces were created around the brains and spinal cords to make them more evident in the 3D reconstructions. Quality thresholds were set based upon visual inspection of the mixed model rendering for both spots and surfaces.

## Immunohistochemistry

Adult animals were perfused with 4% PFA in 1× PBS (15710, Electron Microscopy Sciences), brains, and spinal cords removed, and post-fixed overnight in 4% PFA. Transverse sections of the spinal cord or cortex were embedded in 12% gelatin in 1× PBS (G2500, Sigma-Aldrich, St. Louis, MO) and cut via Vibratome to yield 100 µm sections. Sections were incubated overnight with primary antibodies GFAP (DAKO, Z0334 1:500, RRID:AB_10013482), rinsed, and then incubated for 2 hr with appropriate Alexa Fluor-conjugated secondary antibodies (R37117, Thermo Fisher, Waltham, MA, 1:500). Fluorescent images were acquired using Olympus IX81 or Zeiss 880LSM microscopes.

## Analysis using computational neuroanatomy

We used the BrainGlobe's Initiative software (https://brainglobe.info) of interoperable Python-based tools for the analysis and visualization of the data. For each brain, we captured approximately 500 images. Image planes were captured in a sequence and orientation to maximize compatibility with Brainglobe workflows, specifically in ventral-to-dorsal sequence and with the caudal end of the brain oriented to the left (https://github.com/brainglobe). The 2D images first were assembled in Imaris v9.3.5 and 9.5 (Bitplane, Oxford Instruments). Two channels were created: one to subtract the positive cell signal and generate another set with only background fluorescence. Images were exported to

ImageJ to create a new set of TIFF files. The TIFF files of the sample images were further analyzed with a set of neuroanatomical computational tools developed for analysis of brain serial section imaging using light sheet microscopy. First, we fed both sets of images, background and positive signal images, into the cellfinder, a deep-learning network (residual neural network), to detect the positive cells (https://github.com/brainglobe/cellfinder; *Tyson and Margrie, 2021*) followed by registration and segmentation into a template brain with anatomical annotations based of the Allen Reference Mouse Brain Atlas (https://github.com/brainglobe/brainreg; *Tyson et al., 2022*) and finally visualized with the brainrender (https://github.com/brainglobe/brainrender; *Claudi et al., 2022*).

## Quantification and statistics

Throughout the article, means are used as summary values and standard error of the mean (SEM) as the indicator of variability. Data were tested for assumptions of parametric tests using Kolmogorov–Smirnov and Levene's tests. Values for N, the specific tests and post-hoc analyses, and p-values are provided in the figure legends and in the test of the 'Results' section. All manual quantifications, including behavioral assessment and measurements of lesion size, were performed by blinded observers. Statistical analyses were performed using Prism (GraphPad).

## Acknowledgements

This work was supported by NIH/NINDS R01NS083983 (MB and PT), the Bryon Riesch Paralysis Foundation (MB), The Miami Project to Cure Paralysis (PT), and the Buoniconti fund (PT). We thank Troy Margrie, Adam Tyson, Luigi Petrucco, and others at the Brainglobe initiative, Yania Ondaro-Martinez and Yan Shi at the Miami Project, James Choi for the design of the website, and Vance Lemmon for helpful discussion.

## Additional information

### Funding

| Funder | Grant reference number | Author |
| --- | --- | --- |
| National Institutes of Health | R01NS083983 | Murray G Blackmore |
| The Bryon Riesch Paralysis Foundation | | Murray G Blackmore |
| The Miami Project to Cure Paralysis | | Pantelis Tsoulfas |
| The Buoniconti fund | | Pantelis Tsoulfas |
| State of Florida Red Light Camera Fund | | Pantelis Tsoulfas |

The funders had no role in study design, data collection and interpretation, or the decision to submit the work for publication.

### Author contributions

Zimei Wang, Adam Romanski, Formal analysis, Investigation, Writing - original draft; Vatsal Mehra, Software, Formal analysis, Investigation, Methodology; Yunfang Wang, Software, Investigation, Methodology; Matthew Brannigan, performed and analyzed experiments; Benjamin C Campbell, Gregory A Petsko, Resources, Writing - review and editing; Pantelis Tsoulfas, Conceptualization, Formal analysis, Supervision, Funding acquisition, Investigation, Visualization, Methodology, Project administration, Writing - review and editing; Murray G Blackmore, Conceptualization, Data curation, Formal analysis, Supervision, Funding acquisition, Investigation, Visualization, Writing - original draft, Project administration, Writing - review and editing

### Author ORCIDs

Benjamin C Campbell [iD] http://orcid.org/0000-0001-8041-5561

Gregory A Petsko [ORCID] http://orcid.org/0000-0003-3668-3694
Pantelis Tsoulfas [ORCID] http://orcid.org/0000-0003-1974-6366
Murray G Blackmore [ORCID] http://orcid.org/0000-0001-9345-6688

## Ethics

This study was performed in strict accordance with the recommendations in the Guide for the Care and Use of Laboratory Animals of the National Institutes of Health. All of the animals were handled according to approved institutional animal care and use committee (IACUC) protocols (#4013) of Marquette University. All surgery was performed under ketamine / xylazine anesthesia, and every effort was made to minimize suffering.

## Decision letter and Author response

Decision letter https://doi.org/10.7554/eLife.76254.sa1
Author response https://doi.org/10.7554/eLife.76254.sa2

## Additional files

### Supplementary files

• Transparent reporting form
• Source data 1. Raw cell counts for all animals.
• Source data 2. Raw values for all figures.

### Data availability

All data generated or analysed during this study are included in the manuscript and supporting file or on the associated website 3Dmousebrain.com. Source Data 1 contains complete numerical data from all animals and Source Data 2 contains the numerical data used to generate all figures.

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
