## [Editor Report]

This work is of interest to neuroscientists interested in tissue clearing, viral labeling, and its applications to spinal cord injury in particular, but not exclusively. It provides a significant methodological advance applied for investigating descending pathways from a fundamental perspective and then in the context of potential recovery after a spinal cord injury. This study represents an important new direction and set of tools for the field of motor control.

---

## [Decision Letter]

**Decision letter after peer review:**

Thank you for submitting your article "Brain-wide analysis of the supraspinal connectome reveals anatomical correlates to functional recovery after spinal injury" for consideration by *eLife*. Your article has been reviewed by 3 peer reviewers, and the evaluation has been overseen by a Reviewing Editor and Ronald Calabrese as the Senior Editor. The following individuals involved in review of your submission have agreed to reveal their identity: Julien Bouvier (Reviewer #2); David SK Magnuson (Reviewer #3).

Essential revisions:

Your paper has been examined by three reviewers and myself. Below are their detailed reviews, but please find here a quick summary of their main comments that must be addressed in a revised version in order for us to judge the potential suitability of your study for publication in *eLife*.

(1) Significantly reorganizing the text and improving the figures would help the readers to focus on the most conceptually important parts of the study. Also reviewers found lack of precisions and clarity throughout the manuscript. Please improve.

(2) Dampen the claims that you are the first to use some methods (such as the use of the AAVretro co-injection for example), or that you are providing an experimental platform enabling you to easily perform the same type of analysis.

(3) Quantify the number and localization of the double-labeled cells in the nervous system. Indeed the difference between cervical and lumbar projections might be one of the more interesting aspects of the paper, but quantification and statistics are required here.

(4) Lesions/injuries must be much more precisely described, detailed, explained, justified, quantified (severity assessment) together with their variability and their rationale to be compared to clinical situations.

(5) Improve the way to assess injury/recovery relationships and make sure that the linear regression analysis is the most appropriate and properly performed. We have concerns about statistics and the use of regression analysis, which is quite weak (flawed?) despite being an "accepted" practice.

*Reviewer #1 (Recommendations for the authors):*

I am overall very enthusiastic about the potential that there is in this study, but there are important caveats at this stage. I will therefore support its publication only after strictly all the caveats listed below are thoroughly addressed, including by substantial improvements of the analysis and the illustrations.

1. The manuscript is globally very well written and reads smoothly but it is often hard to "see the forest for the trees" and get a clear take-home message. There are 3 parts, and I consider parts 2 (cervical/lumbar) and 3 (SCI) to be the most informative, conceptually. While acknowledging that the methodology (part 1) is novel and elegant, it uses a substantial number of figures, sometimes for little information. Figure 1 for instance is totally dispensable (at least in the main figures, text paragraphs are also long) since the cell-filling fluorophores aren't used. Btw, the difference in cortical labeling in panels C and D is far from obvious. Figure 3 is also, to me, noise in the story. What is the rationale of injections at the 3 adjacent segments? The reliability of the approach is already convincingly documented by the consistency of findings across animals when injecting the same segment. Plus, this is not the first time AAVs are used to inject the spinal cord with consistent results (see more comments below on this). One could think this figure asks whether the segregation of projection neurons by targeted segment described for cervical and lumbar (part 2) also applies to smaller spatially resolved segments. Do we have to understand that it is not the case? But if that's the aim, co-labeling with the two colors is needed. Overall, I felt distracted by this "unnecessary" data before reaching the hot topic, which starts at Figure 4 (and of course includes Figure 2 for the method). I strongly encourage simplifying and streamlining by reducing the number of dispensable figures and paragraphs. It will help to highlight the real impact of the work which is the labeling and detection method, the differential connectome, and its relevance post-SCI.

2. Coming to the second part. I feel that the topic covered is whether different supra-spinal brain areas control distinct spinal segments through abundant collaterals, or by specific subsets that differ by their projection targets. This matter is becoming the center of attention in many studies on descending tracts. Yet, the present work fails to draw a clear message regarding this, although all the tools and data seem to be there.

(a) My first major concern is the lack of quantification of double-labeled cells from lumbar and cervical injections elsewhere than in the cortex. In the absence of such quantification, Figure 4N is toxic: (i) it conveys the dangerous message that strictly ALL labeled neurons are either green, or red, and thus project to only cervical or only lumbar segments. Is that so? If so, it is a strong message that will deserve substantial discussion; (ii) it may be interpreted that the imaging or detection fails at detecting double-labeled cells – or small cellular contingents -, which would considerably diminish the impact of the whole work. Therefore, the expression and quantification of double-labeled cells in all brain regions MUST be documented in detail, using the SAME imaging pipeline, as it is done for the CST in panel 4P. On panel 4D, yellow labeling is evident in the medulla, but it could just be a visual artifact on the max projection.

(b) Related to this, other labs have shown various degrees of specialization or collateralization in descending tracts using similar strategies, but these works are not cited. Sathyamurthy, et al., (Levine's lab: Cerebellospinal Neurons Regulate Motor Performance and Motor Learning. Cell Rep, 2020) showed exclusive labeling in the cerebellum from combined cervical and lumbar retrograde injections. Usseglio et al., (Bouvier's lab: Control of Orienting Movements and Locomotion by Projection-Defined Subsets of Brainstem V2a Neurons. Curr Biol, 2020) showed specialization in cervical and lumbar projecting neurons in the gigantocellular reticular nucleus. This latter study also identified about 10% of co-labeling. Have the authors here found similar co-labeling in the reticular formation? In general, authors must acknowledge that they are not the firsts to use AAVretro co-injections on descending tracts and must better confront their findings, at the whole-brain scale, to other findings focused on discrete regions when applicable. Likewise, the conclusion that AAVretro is well-suited for terminal-entry and labeling of projection-defined subtypes was already reached by these other publications. Authors could use it to strengthen the validity of their approach (e.g. lines 268, 404, 408/409). Likewise, a recent work by V. Sahni (Corticospinal neuron subpopulation-specific developmental genes prospectively indicate mature segmentally specific axon projection targeting, Cell Reports, 2021) examined differential projections in the CST, which is also relevant here.

(c) At line 266, the co-injection of the two colored viruses in the same segment is claimed to lead to 80% co-labelling, ruling out a preferential expression of one or the other transgene. This is an essential demonstration but neither the quantification nor the data itself are shown. Authors refer to figure 1B/C – they probably mean supplement 1 – but that figure does not show the co-labelling itself at the cellular level nor the quantification. Again, line 411 states "our quantification". This control is made even more important since mGL is way brighter than mSc (btw, this methodological description is also long). This is potentially problematic, and in fact, in Figure 1 supplement 1, more cells are detected in the cortex, pons and medulla with the mGL. This bias must be taken into account: (i) it must be controlled for by co-injecting both viruses in the same segment and quantifying double-labeled cells, (ii) if applicable, it should be highlighted as a potential confounding factor when detecting more cells reaching one segment than the other in the dual tracing experiments of Figure 4. On that note, can it simply be explained by the higher titer of the mGL virus? Ideally, (but I will bear without it if the controls are thoroughly done and show no bias) the same experiments of Figure 4 could be repeated after inverting viruses and/or using the same titers.

(d) On that note also, the choice of the region for inset 2 on Figure 4O is strange: by eye, we see more co-labeling on the periphery of the lumbar-labeled nuclei, but inset 2 magnifies an area with only sparse cervical-projecting cells. Is the co-labeling more massive on the periphery? Where were the counts made to obtain the 1.25 % of co-labeling?

Altogether, it is likely that the second part on the projection patterns to the cervical and lumbar cord will deserve to be unpacked a bit, and making room from the methodology section will help.

3. The spinal cord injury part is very elegant and relevant, but I think it could be brought up sooner in the manuscript. Streamlining some figures before will help. This aspect is in the title but it comes late and is diluted in more "anecdotal" information, so that we are not so receptive anymore to these findings that ARE of great interest. Few things in addition:

(a) I do not understand the rationale of Figure 5 – I follow the authors' arguments, but who doubts that an injured cord would not allow viral entry above the lesion where sensori-motor functions are preserved? Other labs have injected AAVs following SCI (e.g. The Gigantocellular Reticular Nucleus Plays a Significant Role in Locomotor Recovery After Incomplete Spinal Cord Injury. Engmann et al., J Neurosci 2020. PMID: 32978289. See also the work by G. Courtine including ref 23 cited here). These works should also be acknowledged; again, authors aren't the firsts to use AAVs in the spinal cord including after injury. It is also not indicated in the text (line 284) where the injection is done, so one spontaneously thinks it is below the lesion, but we realize when looking at the figure that it is above. To me, this whole section was again noise and I recommend removing it (or moving it to supplemental material) to help highlight the main findings.

(b) It is mentioned that the residual connectivity is the spared connections. But I can't it also relate to regrowth? Isn't there some form of spontaneous regrowth or compensatory branching of severed axons, especially following incomplete contusions? Some work by M. Schwab has suggested this. This should be discussed.

4. In general, there is room for improvements on figures and notably:

(a) Some scale bars are missing. Please check all panels (lesioned cords notably, Figure 6).

(b) Orientation along all axes must be given on all views. Authors alternate sagittal and horizontal views, so it will be helpful to have the orientation into all figure panels.

(c) There is some inconsistency in the orientation: legend of Figure 4 (H-M) refers to a dorsal view, while the same view in Figure 3 is referred to as ventral. Maybe use the term "horizontal" view?

(d) Please include a schematic of the injection timeline and injury when applicable on each figure. Sometimes the schematic is there, but could be more informative (eg., Figure 6 is missing schematic, Figure 5 is missing the timeline, other figures are missing the exact segment targeted).

(e) Figure 6 J to L: we don't see much in these panels. One could make them bigger. In general, the reconstruction panels after the pipeline are barely readable, and I'm sure authors can find a way to improve this.

(f) Names of regions in Figure 2D are not readable, including when zooming on the PDF. Please enlarge/edit.

5. We need more details as to how the moderate injuries were obtained. The methods only speak about mild and presumably severe injuries. Furthermore, how were lesions quantified/classified quantitatively?

6. Please explain how regression slopes were obtained for Figure 7. Likewise, what are the different shades of green in the correlation matrix? This is a highly informative figure, but it is not sufficiently clear how the correlation was made. Maybe also specific that multiple severities of injuries were pooled (if that's the case, I think).

7. For this last one, I will bear without it for acceptance, but it would be icing on the cake. It is very frustrating that authors do not use the 3D imaging to reconstruct and quantify the extent of the spinal lesion, which is instead shown in sections. Other labs have reconstructed contusive injuries in 3D, so it is possible. Including this would strengthen the methodology part and it will substantially help secure the quantification of lesion extent, which as argued before is currently borderline.

*Reviewer #2 (Recommendations for the authors):*

Significant concerns/issues that should be addressed:

There is a lack of precision and clarity as regards a number of important components of the paper that require attention in order to ensure that the reader is not misled. These include clarifying how the data is presented and what the numbers mean. Please see below under clarifications.

The strategy utilized to assess injury/recovery relationships is not the best one, nor even a good one. Following thoracic injuries the recovery should correlate best with spared white matter. Length of injury or extent of astrogliosis, or loss of gray matter volume in thoracic segments are not really relevant to locomotor recovery. If spared white matter (or even better, spared axons at 7 weeks) could be used then a much better correlation could be expected. However, this doesn't detract from the otherwise very interesting set of correlations shown in figure 7B. These are powerful and important results. Please determine if a spared white matter measurement could be made and include correlations with those data. This is particularly important with respect to cervical propriospinal neurons (see comments below on Figure 7 and supplements) and truly getting at "unexplained variability in functional recovery".

Related to this, but an issue that requires separate attention is the highly variable appearance of the injury and injection sites, the variability in the spatial relationship with labeled neurons (lumbar) and how these differences could influence labeling, sprouting of axons of passage and interpretation of the data. Please see comments on Figure 6 and related data below.

Issues/concerns/questions.

Introduction:

You suggest that the spinal cord "relays commands to the periphery through motor and autonomic neurons." but this neglects and underplays the spinal circuitry itself. Processing of the commands is critical for all but a few actions. Please edit to make this clear.

Results:

Figure 1 illustrates an interesting phenomenon in that both labeling strategies resulted in greater variability in CST neurons than in RN and "brainstem". The variability in CST appears to exceed 12% even as the variability in the other populations is very low. Is there an explanation for this or does this represent real biological differences? Does the additional information in figure 1—figure supplement 1 help with this issue?

Did you quantify changes in shape/volume of the brains/subregions of the injured animals? Was registration a problem for the injured animals? This, I think, is part of determining "unexplained variability in functional recovery".

Lines 235-237. In the text for the results shown in Figure 4 you state that cervical label was more abundant, comprising an average of 73% of nuclei throughout the brain, and this is very confusing, suggesting that 73% of the total number of nuclei were labeled by cervical injections. Please re-state this to ensure clarity.

Figure 4O, and Figure 4—figure supplement 1 are way-cool.

In general, the data shown in Figure 4 is important, but again is quite confusing in particular since it does not represent in any way neurons in these regions that were not labeled by injections at either site. Thus, the 100% is again somewhat misleading/confusing. I'm not sure if there is a better way to display the data, but please make these points clear to save the reader some trouble.

Lines 253-255. The data doesn't "use comprehensive, brain-wide approach". Please restate for clarity. You might say that the "data represents a comprehensive, brain-wide approach to both quantify…". Or, "we used a comprehensive, brain-wide approach".

Lines 293. Is the question addressed one of "injury variability" or "recovery variability"? It is unfortunate that the injury model chosen is somewhat less popular than the IH or clip-compression. Presumably the forceps-induced crush injury will have a different pattern of supraspinal survival than a dorsal contusion or clip-compression injury. Please include some discussion of this issue.

Also in this section you chose L4 as your injection site…why? Please provide some rationale and if you would expect that an L2 injection might or might not better reflect supraspinal input that is most important for recovery.

Figure 6 suggests that there is gray matter damage that extends caudal to the injury site with apparent cavitation well apart from T10. Why is the labeling close to the injury in animal 179 so sparse? Overall, these images raise a number of questions and concerns (see below).

Figure 6—figure supplement 1. This figure raises a number of concerns. First of all, the variability between the apparent injection sites and the sites of injury appears to be substantial. An explanation would be helpful. Secondly, the viral injection sites appear to have significant damage. Was this assessed? Could the damage have influenced function, which was not assessed after 7 weeks apparently? Finally, could the damage have influenced sprouting/synaptogenesis in axons of passage that then would have been labeled but would not have been labeled in an uninjured cord? This behooves explanation/exploration.

Also in this figure, the numbers in brackets alongside the animal number do not have an explanation. What do they represent?

Figure 7 and its supplements. This is incredibly important and interesting data. Would it be possible to provide cross-sectional views from caudal, mid and rostral cervical segments to provide a deeper understanding of the laminar locations of these spared propriospinal neurons? Please, sirs?

Discussion:

Line 381. Perhaps it should be information rather than insight, since insight requires some processing and interpretation of information. Insight is used to great effect later in the discussion.

Lines 427-429. SCI definitely influences all sensation, not just pain sensation. Please edit to make this clear.

Line 448. I'm not sure why you say "so-called" neuroanatomical-functional paradox. This is now a well-accepted and important issue that your data speaks to beautifully.

The discussion of lesion size vs spared descending axons is critical, but misses the point referred to above that spared white matter (cross-sectional area) or spared axon counts (individually counted) has been shown to correlate strongly with function after incomplete thoracic SCI. Using "lesion size" is of questionable value as a comparison to the cell counts. The rest of this discussion (lines 458-465) is outstanding. You might state that synaptic silencing and optogenetics are two strategies that could be used to test the true relationship between spared descending pathways/axons and recovery.

Figures:

Figure 1 legend is confusing due to the inconsistent use of capitalized or lowercase and bold. The post-hoc test used is Sidak's.

Figure 7—figure supplement 1 is really confusing. I suspect the circled data points are incorrectly labeled? The numbers of animals with each severity is also incorrectly described.

Methods:

Line 576-577. This sentence does not make sense as written, at least not to me.

---

## [Author Response]

Essential revisions:Your paper has been examined by three reviewers and myself. Below are their detailed reviews, but please find here a quick summary of their main comments that must be addressed in a revised version in order for us to judge the potential suitability of your study for publication in eLife.(1) Significantly reorganizing the text and improving the figures would help the readers to focus on the most conceptually important parts of the study. Also reviewers found lack of precisions and clarity throughout the manuscript. Please improve.

As suggested, we have streamlined the early portions of the manuscript and moved the more technically oriented figures from the main text to the supplementary section. As detailed below we have attempted point-by-point to improve the manuscript in response to each comment on precision and clarity.

(2) Dampen the claims that you are the first to use some methods (such as the use of the AAVretro co-injection for example), or that you are providing an experimental platform enabling you to easily perform the same type of analysis.

We have incorporated existing findings and the use of AAV2-retro throughout the manuscript and have followed suggestions to place our current results in the context of any prior work. We have also modified the text throughout to avoid the implication that we are providing a ready-made platform and instead emphasize that we are providing a framework and example for how such analyses can be performed.

(3) Quantify the number and localization of the double-labeled cells in the nervous system. Indeed the difference between cervical and lumbar projections might be one of the more interesting aspects of the paper, but quantification and statistics are required here.

This represents a major and technically challenging addition to the original manuscript. We agree with the importance of quantification and statistical comparison of double-labeling at the brain-wide scale and emphasize that we would have performed analyses in this fashion had they been possible with the original techniques. Two things prevent this. First, the cellfinder pipeline for identifying and counting cells can be done with one color at the time and does not give the option for colocalized objects. Second, for several reasons the current light sheet system, a first-generation device, is incapable of cellular co-localization at the whole-brain scale. For example, it did not have adjustable light sheet across the entire field of view, resulting in peripheral distortion and preventing precise co-axial placement of two different colors in 10-20 micrometer space. Similarly, the objectives are not corrected for the “longitudinal chromatic aberrations” generated from the refractive index of the immersion media, which vary by wavelength, further complicating two-color colocalization (Schadwinkel H, Selchow O, Weisshart K, Haarstrich J, Birkenbeil J. ZEISS Lightsheet 7: How to Get Best Images with Various Types of Immersion Media and Clearing Agents*.* 2020). Although we expect these issues to improve with newer devices, the present light sheet images are not compatible with colocalization at cellular resolution.

Nevertheless, given the importance of this topic and the strong interest of the reviewers we developed an alternative approach and performed new experiments. This involved high-resolution confocal microscopy applied to 1.5mm brain sections, optically cleared, which was allowed by a recent acquisition of a high speed spinning disk confocal microscope. We emphasize the scale of this effort: approximately 200 hours of imaging, over a million TIFFs and about 10 terabytes of new data collected, and more than 200,000 individual cells detected as expressing one or both fluorophores. Although precise brain-wide registration was not possible, as in the original pipeline, in eleven manually selected regions these data provide comprehensive quantification of retrograde co-labeling from lumbar and cervical levels in four animals, with reference to three control animals that received co-injection to establish expected baseline levels of dual expression. These new data and associated statistics are provided in a new Figure 3 and associated supplements, including a new video to illustrate the technique and main findings.

(4) Lesions/injuries must be much more precisely described, detailed, explained, justified, quantified (severity assessment) together with their variability and their rationale to be compared to clinical situations.

As requested, we have significantly expanded information about the injury method, expanded the rationale for its selection, and have improved our quantification of the lesion severity.

(5) Improve the way to assess injury/recovery relationships and make sure that the linear regression analysis is the most appropriate and properly performed. We have concerns about statistics and the use of regression analysis, which is quite weak (flawed?) despite being an "accepted" practice.

Based on reviewer input and consultation with statisticians we have replaced linear regression with a simplified group-based analysis. Briefly, animals were divided into two groups according to their degree of functional recovery, “high performing” animals with weight-bearing steps and “low-performing” animals without. The number of residual neurons in each brain region was compared between the two with appropriate correction for multiple comparisons. This reworked analysis is presented in a new Figure 6.

Reviewer #1 (Recommendations for the authors):I am overall very enthusiastic about the potential that there is in this study, but there are important caveats at this stage. I will therefore support its publication only after strictly all the caveats listed below are thoroughly addressed, including by substantial improvements of the analysis and the illustrations.1. The manuscript is globally very well written and reads smoothly but it is often hard to "see the forest for the trees" and get a clear take-home message. There are 3 parts, and I consider parts 2 (cervical/lumbar) and 3 (SCI) to be the most informative, conceptually. While acknowledging that the methodology (part 1) is novel and elegant, it uses a substantial number of figures, sometimes for little information. Figure 1 for instance is totally dispensable (at least in the main figures, text paragraphs are also long) since the cell-filling fluorophores aren't used.

We appreciate the input and agree that this figure is more suited to supplementary data. We also shortened the associated description in the main text.

Btw, the difference in cortical labeling in panels C and D is far from obvious.

We agree that the difference in the cortex is not obvious at lower resolution; the overall fluorescence is similar, but in the cytoplasmic case is coming from a fewer number of objects, each one of which is larger. Manual counts do indeed confirm the difference between the two images. We have added higher magnification insets to panels C and D in what is now Figure 1 —figure supplement 1 in C and D, to help illustrate this distinction.

Figure 3 is also, to me, noise in the story. What is the rationale of injections at the 3 adjacent segments? The reliability of the approach is already convincingly documented by the consistency of findings across animals when injecting the same segment. Plus, this is not the first time AAVs are used to inject the spinal cord with consistent results (see more comments below on this). One could think this figure asks whether the segregation of projection neurons by targeted segment described for cervical and lumbar (part 2) also applies to smaller spatially resolved segments. Do we have to understand that it is not the case? But if that's the aim, co-labeling with the two colors is needed. Overall, I felt distracted by this "unnecessary" data before reaching the hot topic, which starts at Figure 4 (and of course includes Figure 2 for the method). I strongly encourage simplifying and streamlining by reducing the number of dispensable figures and paragraphs. It will help to highlight the real impact of the work which is the labeling and detection method, the differential connectome, and its relevance post-SCI.

We now move Figure 3 to Figure 1—figure supplement 5. We do believe the information is important given the interest in the field in performing various injuries across this region of the cord, and the surgical reality that injections don’t always reach their intended spinal level. Indeed, another reviewer raised concerns about apparent variability in our injection locations in later experiments, and the data in this figure offer important reassurance that brain labeling is relatively stable in the face of that variability. Still, we agree that although the data are important for potential users of the approach, the manuscript as a whole benefits from streamlining, and we appreciate the suggestion to move the data to supplementary.

2. Coming to the second part. I feel that the topic covered is whether different supra-spinal brain areas control distinct spinal segments through abundant collaterals, or by specific subsets that differ by their projection targets. This matter is becoming the center of attention in many studies on descending tracts. Yet, the present work fails to draw a clear message regarding this, although all the tools and data seem to be there.(a) My first major concern is the lack of quantification of double-labeled cells from lumbar and cervical injections elsewhere than in the cortex. In the absence of such quantification, Figure 4N is toxic: (i) it conveys the dangerous message that strictly ALL labeled neurons are either green, or red, and thus project to only cervical or only lumbar segments. Is that so? If so, it is a strong message that will deserve substantial discussion;

We apologize for the confusion; this figure was not intended to convey a message of complete segregation of all neurons into distinctly cervical or lumbar-projecting categories. The limitation here is technical. For the reasons described in the introductory remarks above, neither the original registration pipeline nor the imaging system were able to co-localize two colors in individual nuclei on a brain-wide scale. Our previous analysis of the cortex was performed in separate imaging that targeted just that region of the brain at higher resolution, which was facilitated by the ease of identification and relatively shallow position of CST neurons.

In the Prior 4N, we intended to provide an overall index of lumbar versus cervical targeting across various brain regions, a tendency at the level of the population, not colocalization at the individual cell level. To mitigate the confusion, we have removed the two-color scheme in what is now Figure 2N and now simply present bars: the higher the bar, the larger the contribution of lumbar signal to the overall detection in that region. From this it is not possible to extract the precise percent of nuclei that project to cervical, lumbar, or both, but we believe it remains useful to provide an indication, across the whole brain, of the relative targeting. As discussed below we performed an additional experiment to resolve cellular targeting; the distinction between this figure and the next one will further help the reader from over-interpreting the light-sheet based results.

(ii) it may be interpreted that the imaging or detection fails at detecting double-labeled cells – or small cellular contingents -, which would considerably diminish the impact of the whole work. Therefore, the expression and quantification of double-labeled cells in all brain regions MUST be documented in detail, using the SAME imaging pipeline, as it is done for the CST in panel 4P. On panel 4D, yellow labeling is evident in the medulla, but it could just be a visual artifact on the max projection.

We agree with the importance of this question, but as discussed above needed to overcome limitations of the cell detection algorithm and first generation LSFM. The CST, by virtue of its position near the dorsal brain surface, was a special case that allowed higher-resolution LSFM imaging of just that one region. Unfortunately, we can use the main pipeline to count only one wavelength, not automatically count double-labeled nuclei.

As described in detail above we therefore initiated a new set of experiments and an alternative imaging approach based on fast confocal microscopy to provide as much information as possible. This approach allows precise colocalization of nuclei but is not compatible with whole-brain image registration. Still, the new data provide a consistent methodology and high cellular resolution that allowed a quantitative comparison of colocalization in eleven manually identified brain regions, which is arguably the largest-scale effort so far to address cervical/lumbar targeting in multiple supraspinal systems. We appreciate the push to extend our methods in this important new direction.

(b) Related to this, other labs have shown various degrees of specialization or collateralization in descending tracts using similar strategies, but these works are not cited. Sathyamurthy, et al., (Levine's lab: Cerebellospinal Neurons Regulate Motor Performance and Motor Learning. Cell Rep, 2020) showed exclusive labeling in the cerebellum from combined cervical and lumbar retrograde injections. Usseglio et al., (Bouvier's lab: Control of Orienting Movements and Locomotion by Projection-Defined Subsets of Brainstem V2a Neurons. Curr Biol, 2020) showed specialization in cervical and lumbar projecting neurons in the gigantocellular reticular nucleus. This latter study also identified about 10% of co-labeling. Have the authors here found similar co-labeling in the reticular formation? In general, authors must acknowledge that they are not the firsts to use AAVretro co-injections on descending tracts and must better confront their findings, at the whole-brain scale, to other findings focused on discrete regions when applicable. Likewise, the conclusion that AAVretro is well-suited for terminal-entry and labeling of projection-defined subtypes was already reached by these other publications. Authors could use it to strengthen the validity of their approach (e.g. lines 268, 404, 408/409). Likewise, a recent work by V. Sahni (Corticospinal neuron subpopulation-specific developmental genes prospectively indicate mature segmentally specific axon projection targeting, Cell Reports, 2021) examined differential projections in the CST, which is also relevant here.

Thank you for this suggestion and the highly relevant references. We have incorporated them throughout the manuscript in both Results and Discussion. Our results are mostly consistent with this prior work, although we detect somewhat more co-localization, which we discuss. To avoid confusion about rates of co-localization, we let us emphasize here that we focus our analysis specifically on the percent of lumbar-projecting neurons that also take up cervical label. When the Gi data referenced above are reanalyzed in this way, about 20% of lumbar-projecting V2A Gi neurons were found to also take up cervical label, and this is the value we use for comparison in the discussion (i.e. in the prior work colocalized cells are 10% of all Gi neurons but 20% of lumbar-projecting Gi neurons). Overall, we appreciate the strengthening of the manuscript by contextualizing our findings.

(c) At line 266, the co-injection of the two colored viruses in the same segment is claimed to lead to 80% co-labelling, ruling out a preferential expression of one or the other transgene. This is an essential demonstration but neither the quantification nor the data itself are shown. Authors refer to figure 1B/C – they probably mean supplement 1 – but that figure does not show the co-labelling itself at the cellular level nor the quantification. Again, line 411 states "our quantification". This control is made even more important since mGL is way brighter than mSc (btw, this methodological description is also long). This is potentially problematic, and in fact, in Figure 1 supplement 1, more cells are detected in the cortex, pons and medulla with the mGL. This bias must be taken into account: (i) it must be controlled for by co-injecting both viruses in the same segment and quantifying double-labeled cells, (ii) if applicable, it should be highlighted as a potential confounding factor when detecting more cells reaching one segment than the other in the dual tracing experiments of Figure 4. On that note, can it simply be explained by the higher titer of the mGL virus? Ideally, (but I will bear without it if the controls are thoroughly done and show no bias) the same experiments of Figure 4 could be repeated after inverting viruses and/or using the same titers.

Establishing baseline levels of co-expression of the two viruses is indeed essential. We have done so using the same confocal-based approach as the lumbar-cervical experiment, enabling direct comparison. These baseline data are provided in Figure 3 —figure supplement 3. We also attempted the color-swap design suggested above, but many animals in this cohort of animals were lost due to an unfortunate surgical error that led to some leaking of virus directly into brainstem regions, which was discovered only after extensive imaging.

(d) On that note also, the choice of the region for inset 2 on Figure 4O is strange: by eye, we see more co-labeling on the periphery of the lumbar-labeled nuclei, but inset 2 magnifies an area with only sparse cervical-projecting cells. Is the co-labeling more massive on the periphery? Where were the counts made to obtain the 1.25 % of co-labeling?

This question is perceptive but rendered somewhat moot by the replacement of the original quantification with the new confocal-based approach. Cortical co-localization is the focus of new Figure 3 —figure supplement 4 and a video (Figure 3 —figure supplement 5). As we now emphasize in the results and discussion, the original estimate of only 1.25% co-labeling, based on light sheet microscopy, needs to be revised upwards with the new confocal-based data in hand. There is also some nuance, as we show that conclusions about co-localization are highly sensitive to where one sets thresholds of detection. It is important to confirm, however, that both the original and current quantification in the cortex was performed across all of M1 and not only from selected regions.

Altogether, it is likely that the second part on the projection patterns to the cervical and lumbar cord will deserve to be unpacked a bit, and making room from the methodology section will help.

We agree, and appreciate the suggestion to rebalance the manuscript. We view the expanded treatment of cervical/lumbar projection patterns and the addition of the new confocal-based data as a significant improvement to the manuscript – a strong example of how careful peer review can catch inconsistencies and lead to scientific advance.

3. The spinal cord injury part is very elegant and relevant, but I think it could be brought up sooner in the manuscript. Streamlining some figures before will help. This aspect is in the title but it comes late and is diluted in more "anecdotal" information, so that we are not so receptive anymore to these findings that ARE of great interest. Few things in addition:(a) I do not understand the rationale of Figure 5 – I follow the authors' arguments, but who doubts that an injured cord would not allow viral entry above the lesion where sensori-motor functions are preserved? Other labs have injected AAVs following SCI (e.g. The Gigantocellular Reticular Nucleus Plays a Significant Role in Locomotor Recovery After Incomplete Spinal Cord Injury. Engmann et al., J Neurosci 2020. PMID: 32978289. See also the work by G. Courtine including ref 23 cited here). These works should also be acknowledged; again, authors aren't the firsts to use AAVs in the spinal cord including after injury. It is also not indicated in the text (line 284) where the injection is done, so one spontaneously thinks it is below the lesion, but we realize when looking at the figure that it is above. To me, this whole section was again noise and I recommend removing it (or moving it to supplemental material) to help highlight the main findings.

We appreciate the question but would argue that viral uptake by chronically injured neurons is a more open question than the reviewer suggests. A past literature has emphasized neuronal atrophy after injury, raising the possibility that some populations of injured neurons could transport or express the virus less effectively after injury. Conversely, we also considered that because injury triggers a sprouting response we might expect the opposite effect, an increase the efficiency of viral uptake. It is true that virus has been previously used in the chronically injured spinal cord – these references have been added, thank you – but this prior work examined only selected populations and did not attempt to quantify any gains or losses in efficiency after injury. In this context we think a brain-wide, quantitative experiment is something that needed to be done. Our finding that transduction efficiency is unaltered in the chronically injured state is somewhat surprising, and we believe it will be of interest and may serve as a needed touchpoint for preclinical development of this approach. We appreciate the suggestion to clarify the basis for the question and have added text to the Results section better establish the rationale for this experiment.

(b) It is mentioned that the residual connectivity is the spared connections. But I can't it also relate to regrowth? Isn't there some form of spontaneous regrowth or compensatory branching of severed axons, especially following incomplete contusions? Some work by M. Schwab has suggested this. This should be discussed.

This is an important and we appreciate the suggestion to clarify. We have added the following text in the Results section as the experiment is introduced:

“In selecting the L4 location we considered that short-distance sprouting, but not long-distance axon extension, occurs spontaneously after injury ^70,71^. Accordingly, the L4 injection site was chosen to target spared axons while minimizing to the potential for virus to spread rostrally and reach directly injured axons”

4. In general, there is room for improvements on figures and notably:(a) Some scale bars are missing. Please check all panels (lesioned cords notably, Figure 6).

Scale bars have been added, thank you.

(b) Orientation along all axes must be given on all views. Authors alternate sagittal and horizontal views, so it will be helpful to have the orientation into all figure panels.

This was a great suggestion, thank you. Orientation labels have been added to figures or figure legends.

(c) There is some inconsistency in the orientation: legend of Figure 4 (H-M) refers to a dorsal view, while the same view in Figure 3 is referred to as ventral. Maybe use the term "horizontal" view?

We have changed the terminology to “horizontal” throughout.

(d) Please include a schematic of the injection timeline and injury when applicable on each figure. Sometimes the schematic is there, but could be more informative (eg., Figure 6 is missing schematic, Figure 5 is missing the timeline, other figures are missing the exact segment targeted).

Timelines and schematics have been added to the figures that involve spinal injuries.

(e) Figure 6 J to L: we don't see much in these panels. One could make them bigger. In general, the reconstruction panels after the pipeline are barely readable, and I'm sure authors can find a way to improve this.

This is now figure 5. We have simplified, reoriented, and expanded the panels to give readers a better view.

(f) Names of regions in Figure 2D are not readable, including when zooming on the PDF. Please enlarge/edit.

This now figure 1. We replaced the original “69-region” graph with a simplified “25-region” summary graph, which allowed us to increase the size of each name.

5. We need more details as to how the moderate injuries were obtained. The methods only speak about mild and presumably severe injuries. Furthermore, how were lesions quantified/classified quantitatively?

Descriptions of how mild, moderate, and severe injuries were created have been clarified in the Results and in the Methods sections. The method relies on forceps fitted with stoppers that create consistent gaps as they squeeze the spinal cord; the smaller the stopper the narrower the gap and the more severe the injury. Injuries are initially grouped according to this width-defined starting point, but as with all injury models there is residual variability, and a major goal of the manuscript is to employ histological, behavioral, and neuron-counting methods to classify severity. Images of all spinal cords are provided in Figure 5 —figure supplement 1.

6. Please explain how regression slopes were obtained for Figure 7. Likewise, what are the different shades of green in the correlation matrix? This is a highly informative figure, but it is not sufficiently clear how the correlation was made. Maybe also specific that multiple severities of injuries were pooled (if that's the case, I think).

As discussed elsewhere we have reworked this aspect of the manuscript by first grouping animals into two groups based on behavior (low performers, BMS 0-3.5 and high performers, BMS 6-8). We then test for differences in the average number of spared neurons in each region, correcting for multiple comparisons. For reasons also noted by other reviewers, this grouped analysis is more appropriate than linear regression.

7. For this last one, I will bear without it for acceptance, but it would be icing on the cake. It is very frustrating that authors do not use the 3D imaging to reconstruct and quantify the extent of the spinal lesion, which is instead shown in sections. Other labs have reconstructed contusive injuries in 3D, so it is possible. Including this would strengthen the methodology part and it will substantially help secure the quantification of lesion extent, which as argued before is currently borderline.

Regarding 3D imaging of the injury site, we appreciate the suggestion and in hindsight we share the frustration. At the time of the experiment, we were unsure that spinal clearing would be effective (the concern was autofluorescence at the injury site). We felt we couldn’t risk losing the spinal cords – without successful visualization of the injury and injection the experiment is uninterpretable – and so we adopted a conservative approach of conventional tissue processing. Our future work will certainly analyze spinal cords in 3D.

Perhaps the larger point is that we agree with the assessment, shared by reviewer two, that there are limits to the information that can be extracted from the spinal sections in hand, which are not enough to reconstruct the entire injury or – more importantly – to reconstruct spared white matter. Accordingly in the revision we back away from comparing the predictive power of lesion analysis versus our new connectome approach. We offer an update to the spinal cord analyses by measuring as best we can the relative amount of spared tissue (Figure 5 —figure supplement 1) but do so only to provide indexes of severity, not to predict functional outcomes. Overall, in the revised manuscript we try to make clear our position that our new analysis of the residual supraspinal connectome offers an important complement, not a substitute, for careful analysis of the spinal injury.

Reviewer #2 (Recommendations for the authors):Significant concerns/issues that should be addressed:There is a lack of precision and clarity as regards a number of important components of the paper that require attention in order to ensure that the reader is not misled. These include clarifying how the data is presented and what the numbers mean. Please see below under clarifications.The strategy utilized to assess injury/recovery relationships is not the best one, nor even a good one. Following thoracic injuries the recovery should correlate best with spared white matter. Length of injury or extent of astrogliosis, or loss of gray matter volume in thoracic segments are not really relevant to locomotor recovery. If spared white matter (or even better, spared axons at 7 weeks) could be used then a much better correlation could be expected. However, this doesn't detract from the otherwise very interesting set of correlations shown in figure 7B. These are powerful and important results. Please determine if a spared white matter measurement could be made and include correlations with those data. This is particularly important with respect to cervical propriospinal neurons (see comments below on Figure 7 and supplements) and truly getting at "unexplained variability in functional recovery".

To address the technical point first: unfortunately, the four imaged sections from each animal are all that remain (it has been a long road to publication and the experiment ended almost two years ago, during the Covid shutdown) leaving us unable to reconstruct spared white matter. As a partial surrogate we have measured the average width of residual astrocytic bridging and provide those data in Figure 5 —figure supplement 1. Although not as ideal as a whole-spinal cord assessment of residual white matter, we do think this approach captures some information about the relative amount of tissue bridging across injuries and is an improvement over the original analysis of lesion area.

Perhaps the more important point is to back away from the suggestion that we can compare lesion analyses to brain clearing and conclude that brain clearing is more predictive of function. Our assessment of the spinal lesions is not state of the art, which invalidates a head-to-head comparison. It is better to emphasize the prospects of the whole-brain quantification of spared neurons and to position it as a complement, not a substitute, for existing methods that focus on the lesion.

We have revised the manuscript to include the index of astrocytic bridging but have removed any claims of the relative predictive power of the two approaches and have added language to the discussion to emphasize the potentially complementary nature of lesion and whole-brain sparing analyses.

Related to this, but an issue that requires separate attention is the highly variable appearance of the injury and injection sites, the variability in the spatial relationship with labeled neurons (lumbar) and how these differences could influence labeling, sprouting of axons of passage and interpretation of the data. Please see comments on Figure 6 and related data below.

Please see our response under the Figure 6 comments below.

Issues/concerns/questions.Introduction:You suggest that the spinal cord "relays commands to the periphery through motor and autonomic neurons." but this neglects and underplays the spinal circuitry itself. Processing of the commands is critical for all but a few actions. Please edit to make this clear.

Yes, the original version missed the critical role of spinal circuits. We have edited as follows:

“The brain’s control of the body below the head is achieved largely by axonal inputs to spinal circuits, which then interpret and process this information to generate appropriate commands to the periphery through motor and autonomic output neurons.”

Results:Figure 1 illustrates an interesting phenomenon in that both labeling strategies resulted in greater variability in CST neurons than in RN and "brainstem". The variability in CST appears to exceed 12% even as the variability in the other populations is very low. Is there an explanation for this or does this represent real biological differences? Does the additional information in figure 1—figure supplement 1 help with this issue?

We believe the low variability in non-CST neurons in this figure reflects random chance in these four animals, not biological difference. Our reasoning is that in subsequent experiments we observed higher variability across all populations. This can be calculated from the supplemental data (Table 1) and also visually observed in subsequent figures, for example in Figure 1 —figure supplement 2 in panel O in which brainstem populations show similar variability to the CST.

Did you quantify changes in shape/volume of the brains/subregions of the injured animals? Was registration a problem for the injured animals? This, I think, is part of determining "unexplained variability in functional recovery".

This is an interesting question. The premise of registration – the problem it intends to solve – is that individual brains differ in shape due to both inter-animal variation and to potential artifacts during clearing or imaging. During the registration process this variation is eliminated as images are resized allowing for a local, non-linear alignment of the image data to make them conform to a standard size and shape (Niedworok et al., Nature Communications 7, 1-9 2016). Your point is well taken, however: what is gained in standard comparison between animals is lost in the ability to detect changes in the shape of the brain itself. This important point has been added to the “caveat” paragraph of the discussion.

Lines 235-237. In the text for the results shown in Figure 4 you state that cervical label was more abundant, comprising an average of 73% of nuclei throughout the brain, and this is very confusing, suggesting that 73% of the total number of nuclei were labeled by cervical injections. Please re-state this to ensure clarity.

This was confusing as written. This sentence was removed and replaced as part of a larger reorganization based on the addition of new colocalization data. An improved description of how we quantified cervical/lumbar targeting can be found in lines 236-245.

Figure 4O, and Figure 4—figure supplement 1 are way-cool.

We agree, thank you! These are now in Figure 2 —figure supplement 1 and the phenomenon in question can be well appreciated in a new video, Figure 3 —figure supplement 2.

In general, the data shown in Figure 4 is important, but again is quite confusing in particular since it does not represent in any way neurons in these regions that were not labeled by injections at either site. Thus, the 100% is again somewhat misleading/confusing. I'm not sure if there is a better way to display the data, but please make these points clear to save the reader some trouble.

Thank you for this suggestion to improve the display. A very similar concern was raised by reviewer 1 and we provide the common response below:

We apologize for the confusion; this figure was not intended to convey a message of complete segregation of all neurons into distinctly cervical or lumbar-projecting categories. The limitation here is technical. For the reasons described in the introductory remarks above, neither the original registration pipeline nor the imaging system are able to co-localize two colors in individual nuclei on a brain-wide scale. Our previous analysis of the cortex was performed in separate imaging that targeted just that region of the brain at higher resolution, which was facilitated by the ease of identification and relatively shallow position of CST neurons.

In the Prior 4N, we intended to provide an overall index of lumbar versus cervical targeting across various brain regions, a tendency at the level of the population, not colocalization at the individual cell level. To mitigate the confusion, we have removed the two-color scheme in what is now Figure 2N and now simply present bars: the higher the bar, the larger the contribution of lumbar signal to the overall detection in that region. From this it is not possible to extract the precise percent of nuclei that project to cervical, lumbar, or both, but we believe it remains useful to provide an indication, across the whole brain, of the relative targeting. As discussed elsewhere we also performed an additional experiment to resolve cellular targeting; the distinction between this figure and the next one will further help the reader from over-interpreting the light sheet based results.

Lines 253-255. The data doesn't "use comprehensive, brain-wide approach". Please restate for clarity. You might say that the "data represents a comprehensive, brain-wide approach to both quantify…". Or, "we used a comprehensive, brain-wide approach".

Reviewer 1 also objected to this sentence, but for reasons more conceptual. The conclusion has been changed to:

“In summary, these data are broadly supported by current understanding of supraspinal topography while providing region-by-region indexes of lumbar targeting for diverse supraspinal areas.”

Lines 293. Is the question addressed one of "injury variability" or "recovery variability"? It is unfortunate that the injury model chosen is somewhat less popular than the IH or clip-compression. Presumably the forceps-induced crush injury will have a different pattern of supraspinal survival than a dorsal contusion or clip-compression injury. Please include some discussion of this issue.

This is an important point. We have added language to the results explaining our selection of the crush model and have added to the discussion a section on potential differences from a dorsal contusion (for example, we suspect that brainstem tracts in ventral-lateral white matter might fare better after contusion). We agree that applying this method to more clinically relevant contusion models is a critical future direction and are currently seeking funding to obtain an impactor device at Marquette.

Also in this section you chose L4 as your injection site…why? Please provide some rationale and if you would expect that an L2 injection might or might not better reflect supraspinal input that is most important for recovery.

The goal of the experiment is to quantify neurons whose axons survived the injury, which meant it was essential to keep all virus below the injury. Our assumption was that there would be some variability due to both surgical considerations (segmental precision can be harder when returning to a previously injured spinal cord) or to spread of virus from the point of injection. We chose L4 to minimize the chance of virus reaching cut axons above the injury site, which would invalidate the entire approach. We would emphasize the data presented in Figure 1 —figure supplement 5, panel C**,** where we asked how shifting the location of injection affected cell counts in the brain. We found that L1 and L4 injections yielded very similar numbers. Based on this consistent outcome across lumbar injection site we predicted that labeling would be fairly consistent across lumbar injection sites and therefore selected L4, prioritizing the need for injury/injection separation.

We have added language to the Results to explain this rationale.

Figure 6 suggests that there is gray matter damage that extends caudal to the injury site with apparent cavitation well apart from T10. Why is the labeling close to the injury in animal 179 so sparse? Overall, these images raise a number of questions and concerns (see below).Figure 6—figure supplement 1. This figure raises a number of concerns. First of all, the variability between the apparent injection sites and the sites of injury appears to be substantial. An explanation would be helpful. Secondly, the viral injection sites appear to have significant damage. Was this assessed? Could the damage have influenced function, which was not assessed after 7 weeks apparently? Finally, could the damage have influenced sprouting/synaptogenesis in axons of passage that then would have been labeled but would not have been labeled in an uninjured cord? This behooves explanation/exploration.

The injury/injection distance was indeed variable in some animals. We emphasize, however, the data from Figure 1 —figure supplement 5 which shows that retrograde labeling in the brain is fairly constant even as injection sites are shifted across lumbar regions. As shown above, we also offer an analysis that plots estimated injury/injection distance against the total number of labeled neurons in the brain, which detected no correlation. Overall, we concede that there was some surgical variability in the injections but based on our quantitative analyses we do not detect resulting bias in the measurement of residual supraspinal connectivity.

Regarding tissue damage, in many animals this likely occurred during tissue processing on the vibratome. One way to assess whether tearing occurred in vivo or post-mortem is to examine GFAP signal at the edge of the tear, which shows a distinctive elevation only if the damage occurred in vivo. On the other hand, by this same criterium, you are correct that there is genuine damage at injection sites in several animals, but we emphasize that any effects of this damage would be “after the fact” in our experimental design. Injections were done seven weeks after the original injury and all behavioral data were gathered prior to these injections. Importantly, even on the day of injection viral uptake almost certainly outpaces the effects of any damage. The virus spreads a least half a millimeter from the point of injection, likely contacts axon terminals in this zone within minutes, and has a half-life of hours at most. Therefore sprouting or other injury-triggered events near the needle are both too focal and too slow to impact most viral uptake. In summary, the brain labeling we measure is a “snapshot” of axon connectivity as the virus is infused and is unlikely to be impacted by slower-developing damage responses.

Also in this figure, the numbers in brackets alongside the animal number do not have an explanation. What do they represent?

These numbers were BMS scores, reflecting an internal (unsuccessful) effort to search for visual patterns in function/injury relationships. They found their way into the manuscript version by error, and have been removed.

Figure 7 and its supplements. This is incredibly important and interesting data. Would it be possible to provide cross-sectional views from caudal, mid and rostral cervical segments to provide a deeper understanding of the laminar locations of these spared propriospinal neurons? Please, sirs?

This was an outstanding suggestion, and we found that it was indeed possible to extract cross-sectional views from the data. We have now added, for all available animals, cross sections from ~C2, C4-5, and C7 to Figure 6 —figure supplement 2. There is some stretching of nuclei in the Z-plane, but the reader now has access to information regarding the laminar locations of spared propriospinal neurons in all animals.

Discussion:Line 381. Perhaps it should be information rather than insight, since insight requires some processing and interpretation of information. Insight is used to great effect later in the discussion.

Agreed, information is the better term, thank you.

Lines 427-429. SCI definitely influences all sensation, not just pain sensation. Please edit to make this clear.

Thank you for catching this, now fixed by deleting “pain”.

Line 448. I'm not sure why you say "so-called" neuroanatomical-functional paradox. This is now a well-accepted and important issue that your data speaks to beautifully.

Agreed, “so-called” is now deleted. At the risk of triggering philosophical debate, one could take the position that paradoxes don’t truly exist in science, and that what is called a paradox is inevitably the consequence of a flawed premise or incomplete information. We suspect that is the case here, hence “so-called” in the sense that it is now termed a paradox but will be understood eventually. But – we certainly have not proven that point and inserting “so-called” is hardly an effective way to convey that position to the reader. It is distracting here, and we appreciate the suggestion for improvement.

The discussion of lesion size vs spared descending axons is critical, but misses the point referred to above that spared white matter (cross-sectional area) or spared axon counts (individually counted) has been shown to correlate strongly with function after incomplete thoracic SCI. Using "lesion size" is of questionable value as a comparison to the cell counts. The rest of this discussion (lines 458-465) is outstanding. You might state that synaptic silencing and optogenetics are two strategies that could be used to test the true relationship between spared descending pathways/axons and recovery.

This section of the discussion has been substantially rewritten with this comment in mind and now emphasizes the potential synergy between lesion analyses and our current quantification of residual connectivity in the supraspinal connectome. We also incorporated the excellent suggestion to specifically mention chemo- and optogenetics as a needed future direction.

Figures:Figure 1 legend is confusing due to the inconsistent use of capitalized or lowercase and bold. The post-hoc test used is Sidak's.

This is now Figure 1 —figure supplement 1. Capitalization is now standardized and the typo fixed, thank you.

Figure 7—figure supplement 1 is really confusing. I suspect the circled data points are incorrectly labeled? The numbers of animals with each severity is also incorrectly described.

Yes, the labels were incorrect. This figure has been removed and replaced by new Figure 6

Methods:Line 576-577. This sentence does not make sense as written, at least not to me.

This was confusing as written, thank you for catching this. We share the re-write below. Had we known it sooner this one detail would have saved us more than a month of troubleshooting, so we appreciate the suggestion to make it as clear as possible.

“Image planes were captured in a sequence and orientation to maximize compatibility with Brainglobe workflows, specifically in ventral-to-dorsal sequence and with the caudal end of the brain oriented to the left (https://github.com/brainglobe).”